# PERSISTENCE SPHERES: BI-CONTINUOUS REPRESENTATIONS OF PERSISTENCE DIAGRAMS

**Matteo Pegoraro**
Department of Informatics
Università della Svizzera Italiana
Lugano, Switzerland
`{pegorm}@usi.ch`

## ABSTRACT

Persistence spheres are a new functional representation of persistence diagrams. In contrast to existing embeddings such as persistence images, landscapes, or kernel-based methods, persistence spheres define a bi-continuous mapping: they are Lipschitz continuous with respect to the 1-Wasserstein distance and admit a continuous inverse on their image. This provides both stability and geometric fidelity, placing persistence spheres among the few representations of persistence diagrams that offer an inverse-continuity guarantee. We derive explicit formulas for persistence spheres and show that they can be computed efficiently with minimal parallelization overhead. Empirically, we evaluate them on clustering, regression, and classification tasks involving functional data, time series, graphs, meshes, and point clouds. Across these benchmarks, persistence spheres are competitive with, and often improve upon, standard baselines including persistence images, persistence landscapes, persistence splines, and the sliced Wasserstein kernel. Additional simulations in the appendices further support the method and provide practical guidance for tuning its parameters.

## 1 INTRODUCTION

Topological Data Analysis (TDA) is an emerging field that leverages concepts from algebraic topology to study the shape of data, offering coordinate-free and noise-robust methods for extracting meaningful patterns. At the core of TDA lies persistent homology, a framework that captures multi-scale topological features of a dataset. By recording the scales at which features such as connected components, loops, and voids appear (birth) and disappear (death), persistent homology produces compact descriptors of data shape. These descriptors are commonly represented as persistence diagrams (PDs) or barcodes, which provide stable and interpretable summaries amenable to qualitative exploration and (limited) quantitative analysis (Edelsbrunner & Harer, 2010; Oudot, 2015).

**Data Analysis with Persistence Diagrams.** To integrate topological information into data analysis pipelines, PDs are often compared using Wasserstein distances defined through partial optimal transport (POT) (Divol & Lacombe, 2021). These distances play a crucial role in ensuring robustness to perturbations, but they also impose a highly non-linear geometry on the space of PDs. This non-linearity significantly limits the range of statistical tools that can be directly applied to PDs. For instance, even basic operations such as computing averages are non-trivial: they are usually formulated in terms of Wasserstein barycenters (Mileyko et al., 2011), which are computationally intensive to approximate and may fail to yield unique solutions.

**Topological Machine Learning: Vectorizations and Kernel Methods.** To overcome these limitations, numerous vectorization methods have been developed to embed PDs into linear spaces, enabling the use of classical statistical and machine learning techniques. Such embeddings underpin the field of *topological machine learning* (Papamarkou et al., 2024), where topological features and topological loss functions have proven effective in both predictive and representation learning tasks (Moor et al., 2020; Wayland et al., 2024). For comprehensive surveys we refer to Pun et al. (2022); Ali et al. (2023); Papamarkou et al. (2024); here we only recall the main approaches.

Broadly, these methods fall into two main categories. The first consists of explicit embeddings of PDs into linear spaces, while the second comprises kernel methods (Reininghaus et al., 2015; Kusano et al., 2018; Carriere et al., 2017), which employ the *kernel trick* to define feature maps implicitly. Within the class of explicit embeddings, one can further distinguish between approaches based on *descriptive statistics* (Asaad et al., 2022), *algebraic representations* exploiting polynomial rings or tropical coordinates (Kališnik, 2019; Monod et al., 2019; Di Fabio & Ferri, 2015), *functional representations*, which associate to each diagram a scalar field over a chosen domain (Bubenik, 2015; Adams et al., 2017; Biscio & Møller, 2019; Dong et al., 2024; Gotovac Dogaš & Mandarić, 2025) and other approaches (Mitra & Virk, 2024).

**Main Contributions.** In this work, we build on the framework of Gotovac Dogaš & Mandarić (2025) (see Remark 2) and introduce a new functional representation of PDs, mapping each diagram $D$ to a function $\varphi : \mathbb{S}^2 \to \mathbb{R}$. We prove that this map is Lipschitz continuous with respect to the 1-Wasserstein distance between diagrams, and that its inverse, on its domain of definition, is also continuous. The continuity of the forward map guarantees stability, in the sense that similar diagrams produce similar functions, while continuity of the inverse ensures that functional similarity always reflects similarity at the level of diagrams. A key consequence, which we plan to explore in future work, is that this inverse-continuity makes persistence spheres a natural candidate for loss design on persistence diagrams: convergence in function space implies convergence at the diagram level. To the best of our knowledge, similar or stronger guarantees have so far been obtained by restricting to diagrams with at most $n$ points, as in Bate & Garcia Pulido (2024); Mitra & Virk (2024).

## 2 PRELIMINARIES

### 2.1 CONVEX SETS AND SUPPORT FUNCTIONS

We briefly review the notation and concepts from convex analysis and geometry that will be used throughout. Standard references include Rockafellar (1997); Salinetti & Wets (1979).

**Definition 1.** *Given two convex sets $A, B \subset \mathbb{R}^3$, their Minkowski sum and their multiplication with a non-negative scalar $\lambda \geq 0$, are defined as:*

$$A \oplus B = \{a + b \mid a \in A, b \in B\}, \qquad \lambda A = \{\lambda a \mid a \in A\}.$$

**Definition 2.** *Given a compact convex set $A \subset \mathbb{R}^3$, its support function is $h_A : \mathbb{R}^3 \to \mathbb{R}$, defined as $h_A(x) := \max_{a \in A} \langle x, a \rangle$.*

One can check that 1) any support function $h_A$ is completely determined by its restriction on $\mathbb{S}^2$; 2) the operator $A \mapsto h_A$ is linear: $\lambda_1 A \oplus \lambda_2 B \mapsto \lambda_1 h_A + \lambda_2 h_B$.

To compare different convex sets we will use the Hausdorff distance.

**Definition 3.** *Given two compact subsets $A, B \subset Z$, with $(Z, d_Z)$ being a metric space, their Hausdorff distance is defined as $d_H(A, B) := \max\{\max_{a \in A} d_Z(a, B), \max_{b \in B} d_Z(b, A)\}$. With $d_Z(a, B) := \inf_{b \in B} d_Z(a, b)$.*

Now we can state the following classical result.

**Proposition 1.** *Given two compact convex sets $A, B \subset \mathbb{R}^3$, the following holds:*

$$\max_{v \in \mathbb{S}^2} |h_A(v) - h_B(v)| = d_H(A, B),$$

*where $d_H$ is the Hausdorff distance induced by the Euclidean metric on $\mathbb{R}^3$. In particular, the operator $A \mapsto h_A$ is injective.*

### 2.2 INTEGRABLE MEASURES ON $\mathbb{R}^2$

For any Borel measure $\mu$ on $\mathbb{R}^2$, and any $f : \mathbb{R}^2 \to \mathbb{R}$ $\mu$-measurable, we set $\langle \mu, f \rangle := \int_{\mathbb{R}^2} f(p) d\mu(p)$. Moreover, for any $r \geq 0$, we set $B_r = \{p \in \mathbb{R}^2 \mid \|p\|_2 \leq r\}$, and $B_r^c = \mathbb{R}^2 \setminus B_r$.

In the following we will use *integrable* measures and *uniformly* integrable sequences of measures. See Hendrych & Nagy (2022) for more details on such topics.

**Definition 4.** *A positive finite Borel measure on $\mathbb{R}^2$, $\mu$, is called integrable if:*

$$\langle \mu, \| \cdot \|_2 \rangle = \int_{\mathbb{R}^2} \|p\|_2 d\mu(p) < \infty.$$

*Similarly, a sequence of integrable measures $\{\mu_n\}_{n \in \mathbb{N}}$ is uniformly integrable if:*

$$\lim_{r \to \infty} \sup_{n \in \mathbb{N}} \int_{B_r^c} \|p\|_2 d\mu_n(p) = 0.$$

To compare measures, we need weak and vague convergence of measures, which are standard notions in measure theory. See, for instance, Kallenberg (1997).

**Definition 5.** *A sequence of integrable measures $\{\mu_n\}_{n \in \mathbb{N}}$ converges weakly to $\mu$ if $\langle \mu_n, f \rangle \to \langle \mu, f \rangle$ for every $f : \mathbb{R}^2 \to \mathbb{R}$ continuous and bounded. Instead, if $\langle \mu_n, f \rangle \to \langle \mu, f \rangle$ for every $f : \mathbb{R}^2 \to \mathbb{R}$ continuous and compactly supported, we say that $\{\mu_n\}_{n \in \mathbb{N}}$ converges vaguely to $\mu$.*

We write $\mu_n \xrightarrow{w} \mu$ for weak convergence and $\mu_n \xrightarrow{v} \mu$ for vague convergence.

### 2.3 PERSISTENCE DIAGRAMS

For a general overview on PDs and their relevance in TDA, refer to Appendix A. Here, we adopt a measure-theoretic perspective to define PDs. First, we introduce the following notation:

$$\mathbb{R}^2_{x<y} := \{(x,y) \in \mathbb{R}^2 \mid x < y\}, \qquad \Delta := \{(x,y) \in \mathbb{R}^2 \mid x = y\}.$$

**Definition 6.** *A PD is a positive finite measure $\mu_D = \sum_{p \in D} a_p \delta_p$, with $\delta_p$ being the Dirac delta centered in $p \in \mathbb{R}^2$, $D \subset \mathbb{R}^2_{x<y}$ being a finite set, and $a_p \in \mathbb{N}$. We refer to the set $D$ as the support of the diagram.*

Following Divol & Lacombe (2021) we give the following definition.

**Definition 7.** *For any measure $\mu$ and for any subset $Z \subset \mathbb{R}^2_{x<y}$, we define:*

$$\text{Pers}_Z(\mu) = \frac{1}{2} \int_Z (y - x) d\mu((x,y)).$$

*When $Z = \mathbb{R}^2_{x<y}$, we simply write $\text{Pers}(\mu)$.*

As proven in Skraba & Turner (2020), in the context of stability for linear operators defined on spaces of measures, we are forced to work with the 1-Wasserstein metric. To introduce such a metric with a notation convenient for the proofs that follow, we define the following terms.

**Definition 8.** *Consider two diagrams $\mu_D$ and $\mu_{D'}$. A partial matching between $\mu_D = \sum_{p \in D} a_p \delta_p$ and $\mu_{D'} = \sum_{p \in D'} b_p \delta_p$ is a triplet $(D_\gamma, D'_\gamma, \gamma : D_\gamma \to D'_\gamma)$ such that:*

- *$D_\gamma \subset D$ and $D'_\gamma \subset D'$;*

- *$\gamma : D_\gamma \to D'_\gamma$ is a bijection.*

We may indicate a partial matching just with $\gamma$, for the sake of brevity.

Given a partial matching $\gamma$ between $\mu_D = \sum_{p \in D} a_p \delta_p$ and $\mu_{D'} = \sum_{p \in D'} b_p \delta_p$, for every $p \in D_\gamma$, we set $\gamma_p := \min\{a_p, b_{\gamma(p)}\}$. Similarly, for every $q \in D'_\gamma$, we set $\gamma_q := \min\{b_q, a_{\gamma^{-1}(q)}\}$. The cost of $\gamma$ can then be defined as follows:

$$c(\gamma) := \sum_{p \in D_\gamma} \gamma_p \|p - \gamma(p)\|_\infty + \sum_{p \in D_\gamma} (a_p - \gamma_p)\|p - \Delta\|_\infty + \sum_{q \in D'_\gamma} (b_q - \gamma_q)\|q - \Delta\|_\infty +$$
$$\sum_{p \in D \setminus D_\gamma} a_p \|p - \Delta\|_\infty + \sum_{q \in D' \setminus D'_\gamma} b_q \|q - \Delta\|_\infty. \tag{1}$$

**Definition 9.** *The 1-Wasserstein distance between PDs is defined as:*

$$W_1(\mu_D, \mu_{D'}) = \inf\{c(\gamma) \mid \gamma \text{ partial matching between } \mu_D \text{ and } \mu_{D'}\}.$$

**Remark 1.** *The definition of the $1$-Wasserstein distance adopted here is equivalent to other formulations in the literature. In particular, the bijection $\gamma : D_\gamma \to D'_\gamma$ can be interpreted as a transport map between the measures $\sum_{p \in D_\gamma} \gamma_p\, \delta_p$ and $\sum_{q \in D'_\gamma} \gamma_q\, \delta_q$, while the associated cost $c(\gamma)$ corresponds to the transportation cost, including the cost of sending the remaining mass of both diagrams to the diagonal $\Delta$ (see Divol & Lacombe (2021)).*

We recall the following key result from Divol & Lacombe (2021).

**Theorem 1.** $W_1(\mu_D, \mu_{D_n}) \to 0$ *if, and only if,* $\mu_{D_n} \xrightarrow{v} \mu_D$ *and* $\mathrm{Pers}(\mu_{D_n}) \to \mathrm{Pers}(\mu_D)$.

### 2.4 LIFT ZONOIDS OF DISCRETE MEASURES

We now introduce the final components needed to define our topological summaries. Throughout, we adopt the following notation: for a point $p = (x, y) \in \mathbb{R}^2$, we set $(1, p) := (1, x, y) \in \mathbb{R}^3$. The reader may refer to Figure 1 for a visual illustration of the constructions introduced below. For conciseness, Figure 1 also anticipates material from the following sections and therefore includes some notation that will be formally introduced later on.

As a preliminary step, we recall the construction of the lift zonoid associated with an integrable measure, as presented in Koshevoy & Mosler (1998); Hendrych & Nagy (2022). For simplicity and coherence with our setting, we restrict attention to discrete measures.

**Definition 10.** *Given a discrete measure $\mu = \sum_{i=1}^n a_i \delta_{p_i}$, $p_i \in \mathbb{R}^2$ and $a_i > 0$, the lift zonoid of $\mu$ is the following convex set (zonotope):*

$$Z_\mu = \bigoplus_{i=1}^n a_i[0, (1, p_i)] \subset \mathbb{R}^3,$$

*with $[0, (1, p_i)]$ being the segment joining the origin $0 \in \mathbb{R}^3$ and the point $(1, p_i)$. In particular, the lift zonoid of the zero measure is the set $\{0\} \in \mathbb{R}^3$.*

Note that the lift zonoid construction is linear: $\lambda_1 \mu_1 + \lambda_2 \mu_2 \mapsto \lambda_1 Z_{\mu_1} \oplus \lambda_2 Z_{\mu_2}$. Moreover, the support function of $[0, (1, p)]$ amounts to $v \mapsto \mathrm{ReLU}(\langle v, (1, p) \rangle) := \max\{0, \langle v, (1, p) \rangle\}$, as the maximum of the inner product is attained on one of the extremes of the segment. Hence, by linearity, lift zonoids of discrete measures can conveniently be expressed as sums of rectified linear units.

Koshevoy & Mosler (1998); Hendrych & Nagy (2022) prove the following result.

**Proposition 2.** *Given an integrable measure $\mu$ and a sequence of integrable measures $\{\mu_n\}_{n \in \mathbb{N}}$, the following hold:*

$$d_H(Z_\mu, Z_{\mu_n}) \to 0 \text{ if, and only if, } \mu_n \xrightarrow{w} \mu \text{ and } \{\mu_n\}_{n \in \mathbb{N}} \text{ is uniformly integrable.}$$

## 3 PERSISTENCE SPHERES

We are now ready to define persistence spheres as the support functions (see Definition 2) of lift zonoids associated with (weighted) PDs, restricted to $\mathbb{S}^2$. A running example of this construction is provided in Figure 1.

As for other functional representations of PDs, see Adams et al. (2017), we need to re-weight diagrams with a function $\omega : \mathbb{R}^2 \to (0, 1]$ so that the weight assigned to points goes to zero as we approach $\Delta$. Given a diagram $\mu_D = \sum_{p \in D} a_p \delta_p$ and a function $\omega : \mathbb{R}^2 \to (0, 1]$, we set $\mu_D^\omega := \sum_{p \in D} \omega(p) a_p \delta_p$.

**Definition 11.** *Given a PD $\mu_D$ and a function $\omega : \mathbb{R}^2 \to (0, 1]$, the persistence sphere (PS) of $\mu_D$ with weighting $\omega$ is defined as $\varphi_{\mu_D}^\omega := (h_{Z_{\mu_D^\omega}})_{|\mathbb{S}^2}$.*

For any function $\omega : \mathbb{R}^2 \to (0, 1]$ we set $\Gamma_\omega(p) := \omega(p)(1, p)$. We want to control the decay of $\omega$ as points approach the diagonal. We do so with the following technical conditions.

**Definition 12.** *A function $\omega : \mathbb{R}^2 \to (0, 1]$ is called a stable lift weighting if:*

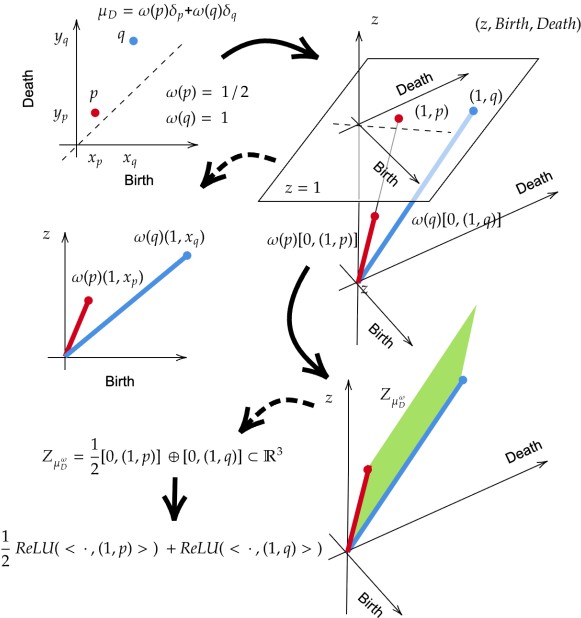

Figure 1: A detailed example illustrating the construction of the lift zonoid of a discrete measure. In the upper left panel, we start from the measure associated with a PD $D$, endowed with a weighting $\omega$ (see Section 3). From the point $p$ we obtain the segment $\omega(p)[0, (1, p)]$, and analogously for $q$, embedded in $\mathbb{R}^3$ with coordinates $(z, \text{Birth}, \text{Death})$. We also provide a 2D representation given by the projection onto the plane $\text{Death} = 0$. The lift zonoid $Z_{D^\omega}$ is then obtained as the Minkowski sum of these two segments. Finally, we report the explicit expression of its support function, which, by linearity, is obtained as the sum of the support functions of each segment.

- $\Gamma_\omega$ is $C$-Lipschitz w.r.t. $\|\cdot\|_2$ for some $C > 0$;

- the following inequality is satisfied for every $p = (x, y) \in \mathbb{R}^2_{x<y}$ and some fixed $C' > 0$:

$$\|\Gamma_\omega(p)\|_2 \leq C' \left( \frac{y - x}{2} \right) = C' \|p - \Delta\|_\infty.$$

In Definition 12, we used the term "lift weighting" to emphasize its role in the context of lift zonoids. Since no ambiguity arises in this work, we will omit the qualifier "lift" from now on for brevity.

**Definition 13.** *A continuous function $\omega : \mathbb{R}^2 \to (0, 1]$ is called an effective (lift) weighting if for any sequence of diagrams $\{\mu_{D_n}\}_{n \in \mathbb{N}}$,*

$$\lim_{r \to \infty} \sup_n \int_{B_r^c} \omega(p) \, \|p\|_2 \, d\mu_{D_n}(p) = 0 \quad \implies \quad \lim_{r \to \infty} \sup_n \text{Pers}_{B_r^c}(\mu_{D_n}) = 0.$$

Definition 13 controls the behavior of $\Gamma_\omega$ at infinity. To see that, note that, for every $\varepsilon > 0$, there is $R$ such that for every $r > R$ and $p \in B_r^c$ we have: $\frac{\|p\|_2}{\|(1,p)\|_2} \geq 1 - \varepsilon$. For any such $r$:

$$(1 - \varepsilon) \sup_n \int_{B_r^c} \omega(p) \|(1, p)\|_2 d\mu_{D_n}(p) \leq \sup_n \int_{B_r^c} \omega(p) \|p\|_2 d\mu_{D_n}(p).$$

Which means that, in the context of the definition, we have:

$$\lim_{r \to \infty} \sup_n \int_{B_r^c} \|\Gamma_\omega(p)\|_2 d\mu_{D_n}(p) \to 0. \tag{2}$$

We now provide examples of stable and effective weightings.

**Proposition 3.** *Set $\lambda(p) := \frac{y-x}{2\|(1,p)\|_2}$. The following are stable weightings:*

$$\widetilde{\omega}(p) = \lambda(p)^\alpha, \qquad \omega_K^\alpha(p) = \frac{2}{\pi} \arctan\left(\frac{\lambda(p)^\alpha}{K^\alpha}\right),$$

*for any $K > 0$ and $\alpha \geq 1$. They are also effective weightings for $\alpha = 1$.*

In what follows, in case $\alpha = 1$, we write $\omega_K$ for simplicity. The weighting $\omega_K^\alpha$ is our preferred choice for constructing PS, because it performs very well in practice (see Section 4). In addition, the parameters $K$ and $\alpha$ are highly interpretable and effective for handling noise in PDs, to the point that they can potentially be qualitatively selected (see Appendix B).

We conclude this section highlighting that, by linearity, the PS of a PD $\mu_D = \sum_{p \in D} a_p \delta_p$, with weighting function $\omega$, can be explicitly written as:

$$\varphi_{\mu_D}^\omega(v) = h_{Z_{\mu_D^\omega}}(v) = \sum_{p \in D} \omega(p) a_p \operatorname{ReLU}(\langle v, (1,p) \rangle). \tag{3}$$

### 3.1 CONTINUITY THEOREMS

We now state our main results, which contain the continuity properties anticipated in the introduction. First we state and prove them in terms of lift zonoids and Hausdorff distances, which simplifies the proofs, and then, using Proposition 1, we derive the bi-continuity of PSs.

**Theorem 2.** *Let $\mu_D, \mu_D'$ be PDs and let $\omega : \mathbb{R}^2 \to (0,1]$ be a stable weighting. We have:*

$$d_H(Z_{\mu_D^\omega}, Z_{\mu_{D'}^\omega}) \leq \sqrt{2} \max\{C, C'\} W_1(\mu_D, \mu_{D'}),$$

*with $C, C' > 0$ being the stability constants of $\omega$ (see Definition 12).*

**Theorem 3.** *Let $\{\mu_{D_n}\}_{n \in \mathbb{N}}$ be a sequence of PDs such that $d_H(Z_{\mu_{D_n}^\omega}, Z_{\mu_D^\omega}) \to 0$, with $\omega : \mathbb{R}^2 \to (0,1]$ being an effective weighting. Then, $W_1(\mu_{D_n}, \mu_D) \to 0$.*

Summarizing the statements of Theorem 2 and Theorem 3, and writing them replacing lift zonoids with PSs, we obtain the following result.

**Corollary 1.** *Within the setting of the previous results, we have:*

- *for every $p \in [1, \infty]$ there exist $C_p > 0$ such that, for every pair of diagrams $\mu_D, \mu_{D'}$, we have $\|\varphi_{\mu_D}^\omega - \varphi_{\mu_{D'}}^\omega\|_p \leq C_p W_1(\mu_D, \mu_{D'})$;*

- *if $\|\varphi_{\mu_D}^\omega - \varphi_{\mu_{D_n}}^\omega\|_\infty \to 0$, then $W_1(\mu_{D_n}, \mu_D) \to 0$.*

As anticipated in the introduction, Corollary 1 establishes a particularly strong link between the Wasserstein geometry of persistence diagrams and their functional representation. For instance, for general embeddings into Hilbert spaces, a bi-Lipschitz embedding of the Wasserstein space is known to be impossible (Carrière & Bauer, 2019). Consistent with this perspective, in Appendix B.2 we present a simulation showing that PS and SW achieve the highest fidelity with respect to the 1-Wasserstein distance.

**Remark 2.** *Gotovac Dogaš & Mandarić (2025) define their functional representation as $\varphi_{\mu_D}^\omega$, with $\omega(p) = y - x$. This weighting is not stable, since for any $C > 0$, $\|\Gamma_\omega(p)\| > C\|p - \Delta\|_\infty$, whenever $\|p\|_2$ is sufficiently large. In particular, the map $\mu_D \mapsto \varphi_{\mu_D}^\omega$ fails to be stable. For instance, let $D_n = \{p_n\} = \{(n^2, n^2 + \frac{1}{n})\}$. Then $Z_{D_n^\omega} = \frac{1}{n}[0, (1, p_n)]$. If $D = \emptyset$, we obtain*

$$W_1(D_n, D) = \frac{1}{n} \xrightarrow{n \to \infty} 0, \qquad d_H(Z_{D^\omega}, Z_{D_n^\omega}) \geq \frac{\sqrt{2}}{n} n^2 \xrightarrow{n \to \infty} \infty.$$

*On the other hand, reasoning as above one can verify that $\omega(p) = y - x$ is effective. Consequently, Theorem 3 holds for the functional representation in Gotovac Dogaš & Mandarić (2025), although this fact is not established in that work. Moreover, in Appendix E we test this weight function showing how it leads to inferior performances.*

# 4 EXPERIMENTS

We evaluated PSs on a range of clustering, regression, and classification case studies, comparing their performance with persistence images (PIs), persistence landscapes (PLs), persistence splines (PSpl) (Dong et al., 2024), the sliced Wasserstein kernel (SWK), and PersLay architectures (for supervised problems with sufficient sample size). For PSs, PIs, PSpl, and PLs we used random forest classifiers and regressors, while SWK was coupled with SVMs. Performance was measured using $R^2$ for regression and accuracy for classification, averaged over 5 independent runs for the "Eyeglasses" case study and 10 runs for all other supervised tasks. Clustering performance was evaluated via the Rand index, averaged over 200 independent repetitions. Computational aspects and runtime simulations are discussed in Appendix C.

## 4.1 DATASETS

**Clustering Case Study** We consider an unsupervised simulation based on a standard functional data analysis (FDA) generative model (Ramsay & Silverman, 2005). We first construct two smooth random functions $f$ and $g$ by cubic-spline interpolation of points in $[0, 1] \times \mathbb{R}$: on a regular grid $0 = x_1 < \cdots < x_{200} = 1$ we sample $y_i^j \sim \mathcal{N}(0, 50^2)$, $i = 1, \ldots, 200$, $j = 1, 2$, independently, and interpolate $\{(x_i, y_i^j)\}_{i=1}^{200}$ to obtain $f$ ($j = 1$) and $g$ ($j = 2$). For a fixed noise level $\sigma > 0$, we generate 50 noisy realizations of each function by sampling uniformly $\{a_i^j\}_{i=1}^{500}$ i.i.d. in $[0, 1]$ and setting $b_i^j = f(a_i^j) + \varepsilon_i^j$, $\varepsilon_i^j \sim \mathcal{N}(0, \sigma^2)$, and analogously for $g$. The noise level takes values $\sigma \in \{10, 15, 30\}$. Each noisy curve is encoded as a 0-dimensional PD, obtained from the sublevel-set filtration of the linear interpolation of the sampled points. For each topological summary, as well as for the 1-Wasserstein and sliced Wasserstein distances, we compute the pairwise distance matrix, perform hierarchical clustering with average linkage, cut the dendrogram into two clusters, and evaluate the partition via the Rand index. The whole pipeline is repeated 200 times for each $\sigma$ and the best performing algorithm is selected via grid-search.

**"Eyeglasses" Case Study** The "Eyeglasses" dataset is a regression case study we designed using the *eyeglasses* generative model from the `scikit-tda` python package (Saul & Tralie, 2019). This model takes two radii as parameters, and a noise variable which was kept equal to 1. The first radius was always set equal to 20, while the second was sampled according to a normal distribution with mean 10 and standard deviation 2.5. We sampled 2000 point clouds and 1-dimensional PDs were obtained from the Vietoris-Rips filtration. For 5 independent runs, we used a $30\% - 70\%$ split between training and test data and threefold cross-validation was used to select hyper-parameters.

**Functional datasets from the `scikit-fda` Package** For the following functional datasets, we used zero-dimensional persistent homology derived from the sublevel set filtration of the functions. Data were split into training and test sets in a $70\%$–$30\%$ ratio, and hyperparameters were selected via threefold cross-validation. All datasets are freely available in the `scikit-fda` Python package (Ramos-Carreño et al., 2024). The datasets "Growth" and "NO$_x$" were smoothed using Nadaraya-Watson kernel smoother with bandwidth 3, chosen by visual inspection.

The "Tecator" dataset (`https://lib.stat.cmu.edu/datasets/tecator`) consists of publicly available measurements collected using the "Tecator Infratec Food and Feed Analyzer". Building on the derivatives of these curves, we explore the same regression problem as in Ferraty & Vieu (2006), trying to regress the fat content of the food samples.

The "NO$_x$" dataset (Febrero et al., 2008) contains hourly measurements of daily nitrogen oxides (NO$_x$) emissions in the Barcelona area. The data is labeled based on whether the emission curve was recorded on a weekday or a weekend, and our goal is thus to reconstruct this labeling through supervised classification.

The "Growth" dataset (Tuddenham & Snyder, 1954), also known as "The Berkeley Growth Study", contains height measurements of girls and boys, recorded yearly between ages 1 and 18. A common approach is to analyze the first derivative of the growth curves to distinguish growth dynamics between boys and girls (Vitelli et al., 2010).

**Datasets from Bandiziol & De Marchi (2024)**  The classification case studies involving the datasets "DYN SYS", "ENZYMES JACC", "POWER", and "SHREC14" were taken from Bandiziol & De Marchi (2024). As in the previous setting, we used a 70%–30% train–test split, with hyperparameters selected via threefold cross-validation. For these datasets, we could directly rely on the PDs associated with the classification tasks, which are publicly available at `https://github.com/cinziabandiziol/persistence_kernels`.

In selecting the problems, we prioritized classification tasks with balanced classes and diversity in data type, including point clouds, graphs, time series, and 3D meshes. We now summarize the considered datasets; further details can be found in Bandiziol & De Marchi (2024).

The dataset "DYN SYS", first introduced in Adams et al. (2017) and referred to as "Orbit Recognition" in Bandiziol & De Marchi (2024), consists of point clouds generated by a one-parameter discrete dynamical system, with the parameter ranging in $\{2.5, 3.5, 4, 4.1, 4.3\}$. The classification task, considered in Adams et al. (2017); Bandiziol & De Marchi (2024) as well as in our work, is to predict the parameter value from the associated point cloud, a problem also studied in Carriere et al. (2017). For each parameter value, 50 independent point clouds were generated, each containing 1000 points with starting positions chosen uniformly at random, yielding a dataset of 250 elements. The PDs computed by Bandiziol & De Marchi (2024) contain only one-dimensional features.

The dataset "ENZYMES JACC" addresses a graph classification problem. Graphs represent protein tertiary structures obtained from the BRENDA enzyme database (`https://www.brenda-enzymes.org/`), and the task is to classify each of the 600 graphs into one of six enzyme classes. Edges were weighted by their Jaccard index, and PDs were computed from the resulting sublevel set filtration, combining both zero- and one-dimensional features.

The dataset "POWER", from the UCR Time Series Classification Archive (`https://www.cs.ucr.edu/~eamonn/time_series_data_2018/`), consists of 1096 time series. The pipeline in this case applied the sliding window embedding (Ravishanker & Chen, 2021), followed by the extraction of zero-, one-, and two-dimensional features, which were then merged into a single diagram for each time series.

Finally, the dataset "SHREC14" (Pickup et al., 2014) is a benchmark for non-rigid 3D shape classification. It contains meshes of human models across 20 poses and 15 body types (e.g., man, woman, child), resulting in 300 total meshes. In Bandiziol & De Marchi (2024), the Heat Kernel Signature (HKS) (Sun et al., 2009; Bronstein & Kokkinos, 2010) was used to extract one-dimensional PDs from the corresponding sublevel set filtrations.

**Datasets "Human Poses" and "McGill 3D Shapes"**  The remaining datasets, "Human Poses" and "McGill 3D Shapes", were obtained from `https://github.com/ctralie/TDALabs/blob/master/3DShapes.ipynb`. The corresponding classification pipelines are documented in the referenced notebook: for the human pose task, a sublevel set filtration of the height function was used, while for the McGill shape classification task, a sublevel set filtration of the HKS was applied. We note that the "McGill 3D Shapes" dataset used here is a subsample of the original version, which is no longer fully accessible online. In both case studies, the train–test split (80%–20%) was imposed by the dataset limitation of having only 10 samples per class.

## 4.2 Parameters Details

Now we describe the parameters used for the vectorization and kernel methods, while we defer the discussion of the employed PersLay architectures to Appendix D.

**Random Forests and SVM Parameters**  We used SVM pipelines with a regularization parameter $C$, chosen from $\{10^{-3}, 10^{-2}, 10^{-1}, 1, 10, 10^2, 10^3, 10^4\}$ and precomputed kernel (SWK). For the Random Forests models the number of estimators trained by each forest was chosen in $\{100, 200\}$. Both were implemented using the `scikit-learn` Python package (Pedregosa et al., 2011).

**Linearization Methods Parameters**  We now summarize the hyperparameters used for each linearization method. For PIs, PSpl and PLs, the range/support parameters were selected by inspecting the full dataset, independently of the training/test split; this introduces a minor inconsistency, which

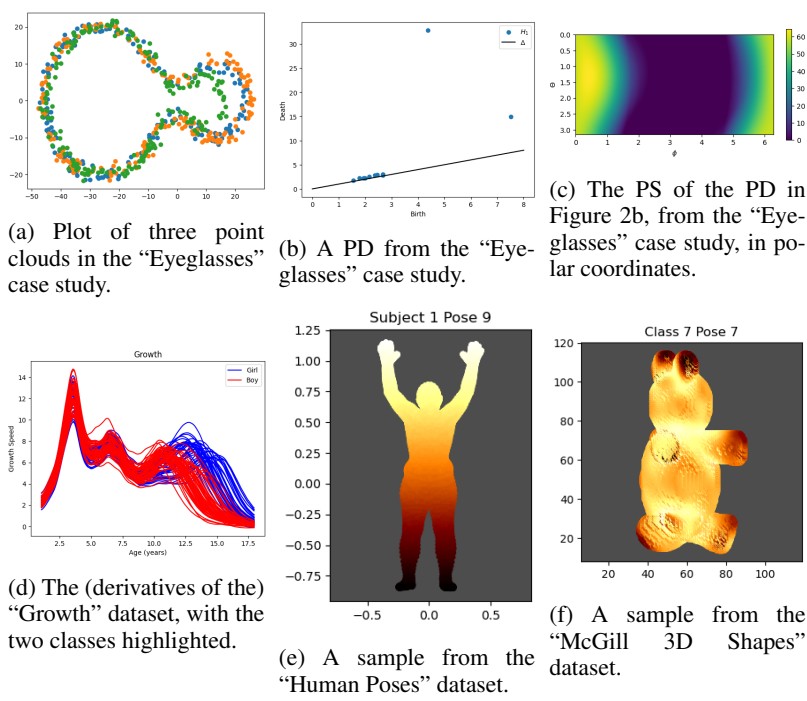

(a) Plot of three point clouds in the "Eyeglasses" case study.

(b) A PD from the "Eyeglasses" case study.

(c) The PS of the PD in Figure 2b, from the "Eyeglasses" case study, in polar coordinates.

(d) The (derivatives of the) "Growth" dataset, with the two classes highlighted.

(e) A sample from the "Human Poses" dataset.

(f) A sample from the "McGill 3D Shapes" dataset.

Figure 2: Data, PDs, and PSs from some of the experiments in Section 4.

could be avoided in practice by choosing sufficiently generous bounds based only on the training data. For PS the domain is fixed and compact, so this issue does not arise.

- PS: we used the weighting function $\omega_K^\alpha$ with $K \in \{0, 10^{-4}, 10^{-3}, 10^{-2}, 10^{-1}, 0.25, 0.5\}$, and $\alpha \in \{1, 3, 5\}$, where $K = 0$ denotes (with a slight abuse of notation) the constant weight $p \mapsto 1$. PSs are functions on $\mathbb{S}^2$ in spherical coordinates and were expanded in spherical harmonics (Müller, 2006) using `pyshtools` (Wieczorek & Meschede, 2018), yielding an orthonormal feature representation for `scikit-learn`. With a Driscoll–Healy grid (Driscoll & Healy, 1994) having $2N_\theta$ latitudinal and $4N_\theta$ longitudinal nodes, the feature dimension is $N_\theta^2/2$; we cross-validated $2N_\theta \in \{30, 40, 50, 60, 70\}$. For the clustering pipeline we used $2N_\theta = 100$, $K \in \{10^{-3}, 10^{-2}, 10^{-1}, 0.3, 0.6\}$ and $\alpha \in \{1, 4, 8\}$. Only for the McGill 3D Shapes dataset, we considered $2N_\theta = 14$.

- PI: with `scikit-tda persim`, we set `pixel_size` by enclosing all PDs in a birth–persistence rectangle and dividing its shortest side by $N_{\text{pix}} \in \{100, 500\}$, then rounding to the nearest power of 10. Using the default Gaussian kernel, we took $\sigma = $ `pixel_size`$/N_\sigma$, $N_\sigma \in \{0.1, 1, 10, 10^2, 10^3, 10^4, 10^5, 10^6\}$, and the persistence exponent $n \in \{1, 2, 4, 8\}$ in `weight_params`. For clustering we restricted to $N_{\text{pix}} \in \{100, 500\}$, $N_\sigma \in \{10^{-3}, 10^{-2}, 10^{-1}, 10^2\}$, and $n \in \{2, 4, 8\}$, and for McGill 3D Shapes, we restricted to $N_{\text{pix}} \in \{5, 10, 20\}$.

- PL: all persistence landscapes were evaluated on a common grid of 5000 points and concatenated (no hyperparameters). For clustering we used a grid of 1000 points.

- PSpl: following (Dong et al., 2024), we used a spline grid of size $h^2$ with $h \in \{5, 10, 20, 40, 50\}$ and iterations in $\{5, 10, 50, 100\}$. As eminence function, we adopted the persistence-based one from the original `matlab` code, ported to `python` from "eminencef.m". For clustering we restricted to $h \in \{10, 20, 40\}$ and iterations in $\{10, 50, 100\}$.

- SWK: we used the sliced Wasserstein kernel from `gudhi` (Project, 2025), fixing $M = 100$ directions and tuning $\sigma \in \{10^{-5}, 10^{-4}, 10^{-3}, 10^{-2}, 10^{-1}, 1, 10\}$ for the Gram matrix.

Table 1: Results of the case studies: we report average $R^2$ for regression and average accuracy for classification, across 5 runs for Eyeglasses and 10 runs for the remaining supervised tasks. Unsupervised clustering (FDA rows) is evaluated via Rand index over 200 runs. We report mean $\pm$ standard deviation. Bold entries denote the best-performing method in each row; a dagger $^\dagger$ marks methods whose 95% confidence interval overlaps with that of the best method.

| | PS | PI | PL | PSpl | PersLay | SWK |
|---|---|---|---|---|---|---|
| **Unsupervised** | | | | | | |
| FDA $\sigma = 10$ ($0.810 \pm 0.221$) | $\mathbf{0.845 \pm 0.158}$ | $0.786 \pm 0.165$ | $0.753 \pm 0.213$ | $0.556 \pm 0.097$ | - | $0.762 \pm 0.220$ |
| FDA $\sigma = 15$ ($0.717 \pm 0.223$) | $\mathbf{0.806 \pm 0.167}$ | $0.730 \pm 0.159$ | $0.676 \pm 0.200$ | $0.538 \pm 0.062$ | - | $0.696 \pm 0.207$ |
| FDA $\sigma = 30$ ($0.548 \pm 0.107$) | $\mathbf{0.688 \pm 0.144}$ | $0.621 \pm 0.103$ | $0.542 \pm 0.085$ | $0.518 \pm 0.014$ | - | $0.578 \pm 0.120$ |
| **Regression** | | | | | | |
| Eyeglasses | $0.966 \pm 0.003^\dagger$ | $0.922 \pm 0.009$ | $0.955 \pm 0.018^\dagger$ | $0.971 \pm 0.011^\dagger$ | $0.248 \pm 0.031$ | $\mathbf{0.971 \pm 0.003}^\dagger$ |
| Tecator | $0.969 \pm 0.009^\dagger$ | $0.900 \pm 0.064$ | $0.954 \pm 0.011^\dagger$ | $\mathbf{0.970 \pm 0.010}^\dagger$ | $0.895 \pm 0.029$ | $0.953 \pm 0.010$ |
| **Classification** | | | | | | |
| Growth | $\mathbf{0.850 \pm 0.052}^\dagger$ | $0.743 \pm 0.135^\dagger$ | $0.768 \pm 0.060$ | $0.807 \pm 0.033^\dagger$ | $0.807 \pm 0.043^\dagger$ | $0.768 \pm 0.058$ |
| NOx | $\mathbf{0.869 \pm 0.041}^\dagger$ | $0.780 \pm 0.060$ | $0.789 \pm 0.062$ | $0.823 \pm 0.033^\dagger$ | $0.717 \pm 0.078$ | $0.840 \pm 0.055^\dagger$ |
| DYN_SYS | $0.829 \pm 0.028^\dagger$ | $0.419 \pm 0.015$ | $\mathbf{0.840 \pm 0.024}^\dagger$ | $0.829 \pm 0.032^\dagger$ | $0.696 \pm 0.044$ | $0.828 \pm 0.028^\dagger$ |
| ENZYMES_JACC | $0.349 \pm 0.036^\dagger$ | $0.342 \pm 0.036^\dagger$ | $\mathbf{0.377 \pm 0.032}^\dagger$ | $0.373 \pm 0.044^\dagger$ | $0.243 \pm 0.023$ | $0.283 \pm 0.055$ |
| POWER | $\mathbf{0.769 \pm 0.021}^\dagger$ | $0.653 \pm 0.066$ | $0.756 \pm 0.018^\dagger$ | $0.748 \pm 0.022^\dagger$ | $0.725 \pm 0.038$ | $0.767 \pm 0.150^\dagger$ |
| SHREC14 | $0.931 \pm 0.022^\dagger$ | $0.894 \pm 0.071^\dagger$ | $0.943 \pm 0.024^\dagger$ | $\mathbf{0.949 \pm 0.023}^\dagger$ | $0.879 \pm 0.018$ | $0.886 \pm 0.092^\dagger$ |
| Human Poses | $\mathbf{0.640 \pm 0.077}^\dagger$ | $0.530 \pm 0.081^\dagger$ | $0.405 \pm 0.106$ | $0.510 \pm 0.102$ | - | $0.345 \pm 0.082$ |
| McGill 3D Shapes | $0.544 \pm 0.085^\dagger$ | $0.461 \pm 0.151$ | $\mathbf{0.678 \pm 0.102}^\dagger$ | $0.561 \pm 0.104^\dagger$ | - | $0.567 \pm 0.130^\dagger$ |

## 4.3 RESULTS

As reported in Table 1, PSs consistently matched or outperformed established topological representations across all the considered tasks. PSplines also performed very well in all supervised settings, suggesting a robust and remarkably low-dimensional representation that is particularly convenient for fitting supervised models; however, in line with Appendix B.2, they performed markedly worse than the other methods in the unsupervised scenario. It is also worth noting that PersLay was likely penalized by the relatively small sample sizes (often between 100 and 1000 observations) and, in the case of Eyeglasses, by the fact that we were unable to identify a network architecture yielding competitive performance (see also Appendix D).

In keeping with the fact that all considered methods have been successfully used in the literature, none of them was dramatically inferior overall. The main practical difficulty arose with PIs, for which identifying suitable parameter ranges proved more delicate and occasionally led to very long runtimes due to slow training of random forests (and other supervised models we tried). Finally, for the McGill 3D Shapes dataset we observed that PSs and, in particular, PIs were more unstable and harder to optimize, with higher variability in accuracy compared to the other methods; in response, we substantially reduced the dimensionality of their vectorizations. We did not observe similar behavior on any of the other datasets, as further illustrated in Appendix E.

## 5 CONCLUSION AND BROADER IMPACT

We introduced PSs, a novel functional representation of persistence diagrams that is both Lipschitz continuous and admits a continuous inverse on its image, yielding a bi-continuous correspondence with respect to the 1-Wasserstein geometry. This combination of stability and geometric fidelity sets PSs apart from existing vectorization methods. Empirically, we find that PSs are not only competitive with, but frequently outperform, widely used alternatives such as PIs, PLs, and SWK.

Several avenues for future work remain. Alternative weighting schemes may yield more expressive summaries. Tools from FDA could support advanced statistical methodologies, such as confidence sets, hypothesis testing, and limit theorems for point processes Biscio et al. (2020). Reconstruction techniques for recovering PDs from scalar fields on the sphere are under development, exploiting existing optimization frameworks for ReLU neural networks. Visualization strategies could enhance interpretability. PSs may also be integrated into modern representation-learning pipelines through differentiable topological layers, for instance in a PersLay-compatible fashion, thereby broadening their applicability. Finally, extending the construction to signed measures could provide a natural vectorization for bi-parameter persistence Loiseaux et al. (2023).

## REPRODUCIBILITY STATEMENT

Section 4.2 and Appendix C provide the main details required to reproduce our results. All datasets used are publicly available, and the explicit formulation of our method in Section 3 ensures reproducibility. We also submit the code necessary to run the experiments as supplementary material (excluding the data).

## THE USE OF LARGE LANGUAGE MODELS

Large Language Models were occasionally employed to refine and polish the writing.

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

## A  FILTRATIONS AND PERSISTENCE DIAGRAMS

As mentioned in the introduction, TDA provides a wide range of techniques to extract features with desirable invariance properties. It does so by leveraging algebraic topology, which offers a natural framework for identifying structures that remain unchanged under continuous deformations of the domain (Hatcher, 2000). A central tool in TDA is persistent homology, which builds on these ideas to track the evolution of homological features, such as path-connected components (0-dimensional holes) and loops (1-dimensional holes), across a filtration, that is, a nested family of topological spaces.

Filtrations generated by real-valued functions and point clouds are among the most general and widely used. Given a topological space $X$ and a function $f : X \to \mathbb{R}$, one considers the sublevel sets $X_t = f^{-1}((-\infty, t])$. The changing topology of the family $\{X_t\}_{t \in \mathbb{R}}$ encodes information about the structure of $f$.

Similarly, for a finite set $X \subset \mathbb{R}^n$, one can consider the filtration

$$X_t = \bigcup_{x \in X} \{p \in \mathbb{R}^n : \|x - p\| < t\}.$$

A visual representation of this filtration, known as the Čech filtration of $X$, is provided in Figure 3.

To extract topological information from a filtration, one typically applies homology functors $H_0, H_1, \ldots$ with coefficients in a field. The resulting families of vector spaces, usually referred to as persistence modules, track the birth and death of features such as path-connected components and loops.

To make this information amenable to data analysis, persistence modules are encoded by topological summaries. Among these, persistence diagrams are arguably the most widely used in TDA; for a detailed survey, see, for instance, (Edelsbrunner & Harer, 2008).

Loosely speaking, a persistence diagram is a multiset of points $(c_x, c_y)$ in the upper half of the plane, with $c_y > c_x$, where $c_x$ denotes the value of the parameter $t$ at which a homology class in $X_t$ first appears (its *birth*), and $c_y$ is the value of $t$ at which the same class either disappears or merges with a previously born class (its *death*).

## B  THE ROLE OF THE WEIGHTING FUNCTION AND THE ASSOCIATED PARAMETERS IN PSS

In this section, we examine the weighting function

$$\omega_K(p) = \frac{2}{\pi} \arctan\left(\frac{\lambda(p)^\alpha}{K^\alpha}\right),$$

which is the one used in our simulations and case study, and we analyze the effect of its parameters $K$ and $\alpha$. Observe that studying this function on $\mathbb{R}_{x \leq y}$ is, by symmetry, equivalent to extending it to $\mathbb{R}^2$ by replacing $y - x$ with $|y - x|$. We will adopt this viewpoint throughout the section.

We begin with an informal, qualitative description, aimed at providing an intuitive overview of the roles played by $\omega_K$, $K$, and $\alpha$, and then move on to a rigorous mathematical justification of the claims. The reader may find it helpful to refer to Figure 4 and Figure 5 throughout the discussion.

At a high level, the function $\omega_K$ is a smooth step function on the plane: it starts from $0$ along the diagonal $y = x$ and, as the direction of $p$ rotates away from the diagonal $y = x$ toward $y = -x$, it transitions monotonically to higher values, bounded by $1$. The parameter $K$ controls both the location and the width of this transition region: it determines where the weight assigned to points begins to decrease from $1$ towards smaller values as their persistence approaches $0$, and how spread out this transition is. By contrast, the parameter $\alpha$ primarily affects the steepness of the step, without shifting its location (see Figure 4).

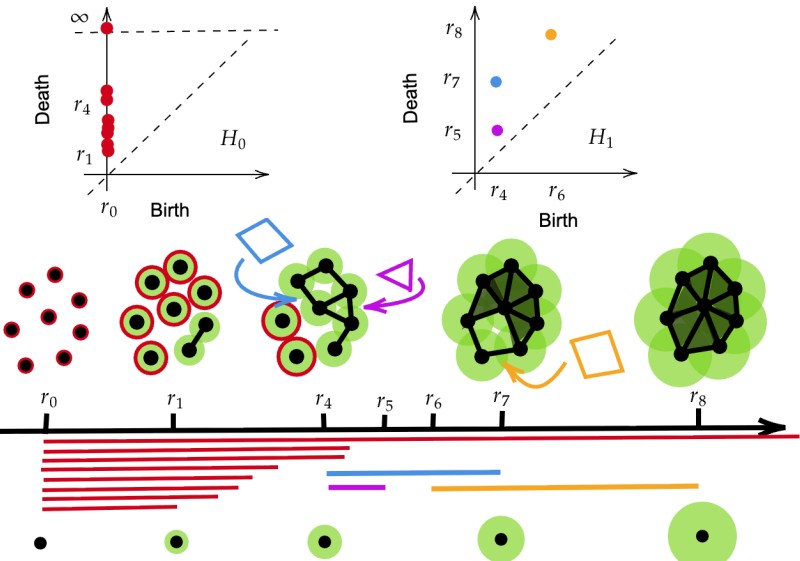

Figure 3: A schematic illustration of a point cloud in $\mathbb{R}^2$, its Čech filtration, and the associated persistence diagrams in homology dimensions 0 and 1. Path-connected components consisting of singletons are highlighted with red circles, while 1-cycles are emphasized using colored polygons. The colors of the polygons are consistent with those of the corresponding points in the diagram associated with $H_1$ and with the horizontal bars representing the life-spans of topological features. Red bars are path connected components.

Recall that increasing $\alpha$ beyond 1 generally breaks the bi-continuity guarantees of the PS representation. However, as we will see in Appendix B.1, choosing $\alpha > 1$ can still be beneficial in practice when working with particularly noisy diagrams. For the rest of this section, we focus on the case $\alpha = 1$.

We now make the above qualitative picture more precise, trying to derive an explicit characterization of how $K$ affects the step function. In doing so, a central role is played by the quantity

$$\lambda(p) = \frac{y-x}{2\|(1,p)\|_2},$$

which, for each point $p = (x,y)$, measures the ratio between its persistence and the Euclidean norm of the corresponding vector $(1,p)$ in the lift-zonoid representation (in the unweighted case). Alternatively, one can see:

$$\lambda(p) = \frac{y-x}{\sqrt{2}\|(0,p)\|_2} \cdot \frac{\|(0,p)\|_2}{\sqrt{2}\|(1,p)\|_2}.$$

The term $\frac{y-x}{\sqrt{2}\|p\|_2}$ is the cosine of the angle between the vectors $(x,y)$ and $(-1,1)$, while the second term is constant if we keep $\|(p)\|_2$ constant. As a consequence, $\lambda$ vanishes on the diagonal $y = x$, its values increase with the angle between $(x,y)$ and the vector $(1,1)$, and they are maximized along $y = -x$, while remaining strictly smaller than 1. Feeding this function into a sigmoidal nonlinearity such as the `arctan` produces a step-like function that is 0 on $y = x$, increases as we move away from the diagonal, and gradually flattens out as we approach $y = -x$, as in Figure 4.

Multiplying $\lambda$ by $1/K$ rescales its Lipschitz constant by $1/K$. Since $|\arctan'(t)| \leq 1$ for all $t$, it follows that, for $\alpha = 1$,

$$\text{Lip}(\omega_K) \leq \frac{2}{\pi} \text{Lip}(\lambda/K) = \frac{2}{\pi} \frac{\text{Lip}(\lambda)}{K}.$$

Hence larger values of $K$ produce a flatter transition (smaller Lipschitz constant), while smaller $K$ yield a sharper step. In other words, $K$ controls the steepness of the resulting step function and

shrinks or enlarges the transition region between the diagonal $y = x$ and the plateau where the weight is close to 1 (see Figure 4).

As shown in Figure 4, the level sets of the weighting are not parallel to the diagonal, which would be the case if $\omega_K$ depended only on the persistence of the points. Instead, they exhibit a radial behavior with respect to the origin. This feature is implicitly enforced by the definition of the lift zonoid, which is built from segments of the form $[0, (1, p)]$, and is somewhat analogous to what happens with the sliced Wasserstein kernel, where points are projected onto straight lines passing through the origin. In fact, both representations induce metrics which are not translation invariant on PDs.

Building on this picture, we now add a further layer of mathematical rigor by deriving explicit expressions for the (asymptotes of the) level sets of $\lambda/K$. These formulas underpin the visualizations reported in Figure 4 and Figure 5, and the practical considerations we discuss in Remark 3.

Since $y \geq x$, for a fixed value $z$ we can write

$$0 \leq z = \frac{\lambda(p)}{K} = \frac{y - x}{2K\|(1, x, y)\|_2}, \qquad (y - x)^2 = (2Kz)^2\big(1 + x^2 + y^2\big).$$

This identity leads to

$$y^2\big((2Kz)^2 - 1\big) + x^2\big((2Kz)^2 - 1\big) + 2xy + (2Kz)^2 = 0. \tag{4}$$

Suppose momentarily that $(2Kz)^2 - 1 \neq 0$. Since $\lambda$ is maximized along the direction $y = -x$, from $\|(1, x, y)\|_2 > \|(x, y)\|_2$ we obtain

$$z < \frac{1}{\sqrt{2}\,K}, \qquad (2Kz)^2 \leq 2.$$

As a consequence, $((2Kz)^2 - 1)^2 - 1 \leq 0$. Using this, one can show that Equation (4) describes a hyperbola, centered at the origin, whose focal points lie on the line $y = -x$ (indeed, $y = x$ corresponds to $z = 0$ and thus cannot be intersected by the hyperbola when $z > 0$). This means that we can interpret the level sets of $\lambda/K$ via the asymptotes of this hyperbola.

Introducing the notation $A = 2Kz$ and $B = 1 - A^2$, the slope of the asymptotes of this hyperbola can be written as

$$m_{1,2} = \frac{1 \pm \sqrt{1 - B^2}}{B}. \tag{5}$$

Recall that $A^2 \in [0, 2]$ and $B \in [-1, 1]$.

Moreover, $B$ is a monotone decreasing function w.r.t. $K$ (and so $1 - B^2$ is monotone increasing) and changes sign at $K = \frac{1}{2z}$.

To get an even more interpretable view on this, let $\theta_i$ be the angle between the vectors $(1, m_i)$ (representing the asymptotes) and $(1, 1)$ (representing $y = x$). That is:

$$\theta_i = \cos^{-1}\left(\frac{1 + m_i}{\sqrt{2(1 + m_i^2)}}\right) \geq 0.$$

Figure 5 displays the functions $\theta_i(K)$, whose behavior we now briefly discuss. As $K$ grows, the lines identified by each angle $\theta_i$ move from the line $y = x$ to the line $y = -x$, with the line of slope $m_1$ rotating counterclockwise and that of slope $m_2$ rotating clockwise. At first sight, Figure 5a may seem to contradict this interpretation for larger values of $K$, as $\theta_1$ suddenly decreases, but a closer inspection shows that this is not the case.

When $B$ changes sign, also $m_{1,2}$ change sign. Moreover, when $K \to \frac{1}{2z}$, $m_1 \to \infty$ and so, when $m_1$ changes sign, it goes from pointing upward almost vertically, to pointing downward almost vertically, and so the angle $\theta_1$ jumps, as represented in Figure 5a, but the line it represents still moves counterclockwise, and our interpretation remains consistent. Instead, since $m_2 \to 0$, it does not jump when $K$ goes across $\frac{1}{2z}$, as shown in Figure 5b.

To summarize, $K$ governs the geometry of the level sets $\omega_K(p) = z$, i.e. the loci of points in the diagram that are assigned a fixed weight $z$. For instance, as $K \to 0$, we have $m_{1,2} \to 1$, meaning that the corresponding level sets move toward the diagonal $y = x$. Conversely, as $K$ increases, the level sets associated with a fixed value $z$ move farther away from the diagonal. The case $A = 1$ (and $B = 0$) yields a singularity, which now admits a clear interpretation: the asymptote with coefficient $m_1$ becomes the vertical line $x = 0$. Indeed, substituting $2Kz = 1$ into Equation (4) gives $2xy + 1 = 0$.

Figure 5 also provides additional insight, with practical implications, into the behavior of $\theta_{1,2}(K)$, which quantifies how quickly the asymptotes are displaced as $K$ varies. The region most relevant for applications is typically the one close to the diagonal (i.e., for smaller values of $K$), where the dependence on $K$ is essentially linear.

For example, and as illustrated in Figure 5b, setting $K = 0.5$ roughly corresponds to assigning a score of $0.5 = \frac{2}{\pi} \arctan(1)$ to points lying at an angle of $\pi/4$ with respect to $y = x$, that is, at an angle of $\pi/2$ with the $x$-axis (namely, on the $y$-axis, halfway between $y = x$ and $y = -x$). By approximate linearity, choosing $K = 0.25$ instead assigns the same score $0.5 = \frac{2}{\pi} \arctan(1)$ to points near an angle of $\pi/8$ with $y = x$, and so on.

In other words, for $K \leq 0.5$, the relationship between $K$ and the angle can be bounded as $\sqrt{2}zK \leq \theta_i \leq \frac{\pi}{2}K$. In particular, $\sqrt{2}zK$ is a first-order approximation of $\theta_{1,2}(K)$ for $K$ close to zero. Before proving this, we summarize these considerations and their practical implications in the following remark.

**Remark 3.** *Taken together, these observations show that any regular grid in $(0, 0.5]$ will move the level set $\omega_K(p) = 0.5$ from arbitrarily close to the diagonal $y = x$ towards the $y$-axis in roughly uniform angular steps. Since noise in persistence diagrams typically lies near $y = x$, it is often preferable, as we do in practice, to instead use an irregular grid that is denser near the lower end of this interval. Additional values larger than $0.5$ can also be considered, especially for unsupervised analyses, if the data are concentrated in the quadrant $x < 0$, $y > 0$ (see Figure 4b) and are strongly affected by noise; in that case, values of $\alpha > 1$ should be explored as well, as illustrated in Appendix B.1, trading bicontinuity guarantees for increased stability. Note, however, that $K = 1/\sqrt{2} \approx 0.7$ corresponds to the level set $\omega_K(p) = 0.5$ lying on the diagonal $y = -x$. We emphasize that this entire analysis is driven solely by the structure of $\lambda/K$ and does not depend on the specific choice of the sigmoidal nonlinearity. Consequently, the qualitative behavior of the level sets and their dependence on $K$ would carry over to other sigmoid functions beyond $\arctan(z)$.*

Lastly we derive the first order approximation of $\theta_{1,2}(K)$ for $K \to 0$.

By Taylor:

$$X = \cos(\theta) = 1 - \frac{\theta^2}{2} + O(\theta^4) \text{ for } \theta \approx 0.$$

Thus, we can write $\theta^2 \approx 2 - 2X$ for $X \approx 1$. Set:

$$\cos(\theta_i) = X_i = \frac{1 + m_i}{\sqrt{2(1 + m_i^2)}}$$

For $K \to 0$, we have $X_i \to 1$. Again by Taylor:

$$2 - 2X_i = \frac{(m_i - 1)^2}{4} + O((m_i - 1)^3) \text{ for } m_i \to 1.$$

Lastly, consider:

$$\frac{(m_i - 1)^2}{4} = \frac{1}{4}\left(\frac{1 \pm \sqrt{1 - B^2}}{B} - 1\right)^2 = \frac{1}{4}\left(\frac{1 \pm A\sqrt{2 - A^2}}{1 - A^2} - 1\right)^2 = \frac{A^2}{2} + O(A^3),$$

for $A \to 0$. Remember that $A = 2Kz$, so if $K \to 0$ then $A \to 0$.

Putting the pieces together, we have:

$$\theta_i^2 \approx \frac{(2Kz)^2}{2}, \text{ for } K \approx 0.$$

Since we know that $\theta_i \geq 0$, we obtain: $\theta_i \approx \sqrt{2}zK$ for $K \approx 0$.

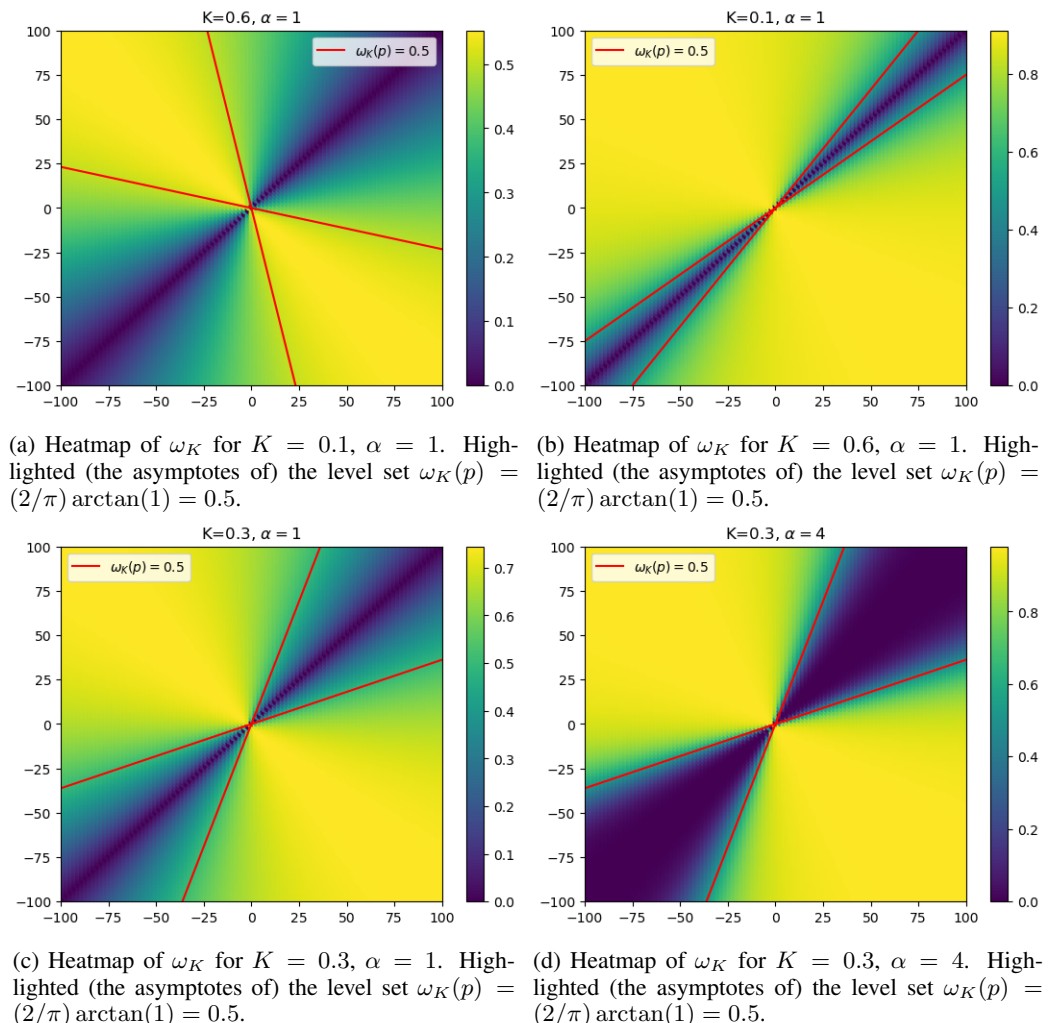

(a) Heatmap of $\omega_K$ for $K = 0.1$, $\alpha = 1$. Highlighted (the asymptotes of) the level set $\omega_K(p) = (2/\pi)\arctan(1) = 0.5$.

(b) Heatmap of $\omega_K$ for $K = 0.6$, $\alpha = 1$. Highlighted (the asymptotes of) the level set $\omega_K(p) = (2/\pi)\arctan(1) = 0.5$.

(c) Heatmap of $\omega_K$ for $K = 0.3$, $\alpha = 1$. Highlighted (the asymptotes of) the level set $\omega_K(p) = (2/\pi)\arctan(1) = 0.5$.

(d) Heatmap of $\omega_K$ for $K = 0.3$, $\alpha = 4$. Highlighted (the asymptotes of) the level set $\omega_K(p) = (2/\pi)\arctan(1) = 0.5$.

Figure 4: Comparison of the step function $\omega_K$ for different values of $K$ and $\alpha$. In the first row, decreasing $K$ shrinks toward the diagonal $y = x$ the region where the weight is significantly reduced for low-persistence points, thereby enlarging the area where the weighting flattens out near 1. The asymptotes, shown for the level set $\omega_K(p) = 0.5$, highlight how the location of a fixed score shifts under different values of $K$. Moreover, the transition from lower to higher scores becomes more gradual for larger $K$, reflecting a smaller Lipschitz constant. In the second row, the role of $\alpha$ in handling highly noisy diagrams is illustrated: increasing $\alpha$ leaves the level set $\omega_K(p) = 0.5$ unchanged in position, but sharply increases the steepness of the step. This produces an extended flat region with low scores near the diagonal, and a correspondingly large flat region with high scores farther away from it.

## B.1  THE ROLE OF $\alpha$

To illustrate the effect of $\alpha$ and its potential relevance in practice, we propose the following simulation study. The guiding idea is that, when persistence diagrams contain an overwhelming number of points close to the diagonal that can be regarded as noise (i.e. do not carry meaningful variability), the 1-Wasserstein distance can be severely affected. Indeed, among the $p$-Wasserstein distances, the case $p = 1$ is the least robust, since the cost of sending points to the diagonal scales linearly with persistence. On the other hand, precisely this linear sensitivity can make the 1-Wasserstein distance particularly effective for discrimination tasks.

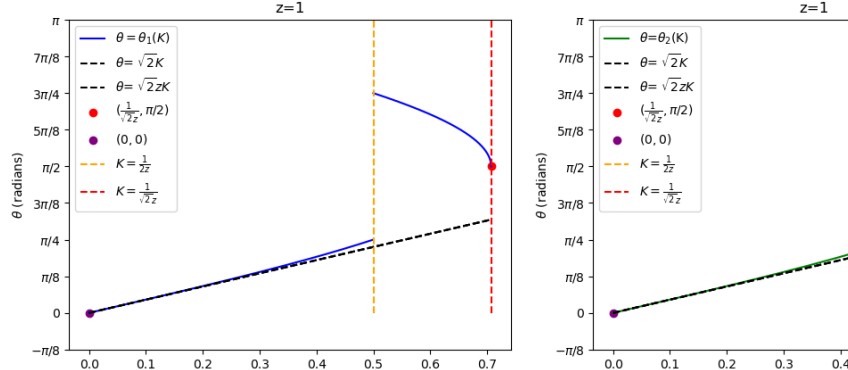

(a) The function $\theta_1(K)$ for $z = 1$, measured in radians. The highlighted points correspond to the two diagonals: $y = x$ (red) and $y = -x$ (purple). The black dashed lines emphasize the approximately linear behavior of $\theta_1(K)$ for small values of $K$, while the vertical orange dashed line marks $K = 1/(2z)$, where $m_1$ becomes singular and the corresponding level set acquires a vertical asymptote.

(b) The function $\theta_2(K)$ for $z = 1$, measured in radians. The highlighted points correspond to the two diagonals: $y = x$ (red) and $y = -x$ (purple). The black dashed lines emphasize the approximately linear behavior of $\theta_{12}(K)$ for small values of $K$, while the vertical orange dashed line marks $K = 1/(2z)$, where $m_1$ becomes singular and the corresponding level set acquires a vertical asymptote.

Figure 5: Behavior of the angle (in radians) of the asymptotes to the level sets as a function of $K$. Near the diagonal $y = x$ (slope $\pi/4$), the angular coefficient varies approximately linearly with $K$, and then starts to increase more rapidly as $K$ approaches $1/(\sqrt{2}\,z)$. The region close to 0 (i.e. near the diagonal) is the most relevant for handling noise, and the essentially linear dependence on $K$ there implies that a simple uniform grid in $K$ explores uniformly the corresponding range of angles.

To mitigate the influence of near-diagonal noise, other linearization methods such as PIs and persistence splines adopt more rapidly decaying weighting schemes. For instance, in the case of PIs one may use higher powers of persistence as weights. This is exactly the role played by $\alpha$ for PSs: as shown in Figure 4d, larger values of $\alpha$ induce a sharper decay of the weighting as one approaches the diagonal $y = x$.

We now present a simple simulation to demonstrate this effect in practice. We generate persistence diagrams belonging to two classes $i = 1, 2$, each of cardinality 50, such that within each class there is a comparable "core" of points away from the diagonal, together with a highly variable number of points near $y = x$. Formally, for each class $i \in \{1, 2\}$ and each replicate $k \in \{1, \ldots, 50\}$, the diagram has the form

$$D_i^k = \{(b_j^{i,k}, d_j^{i,k})\}_{j=1}^{n_k} \cup \{(\hat{b}_r^{i,k}, \hat{d}_r^{i,k})\}_{r=1}^{m_k},$$

and is sampled as follows:

$$
\begin{aligned}
\widetilde{n}_k &\sim \mathcal{N}(N, 5^2), & \widetilde{m}_k &\sim \mathcal{U}([1, N'-1]), & & \\
n_k &= \lfloor |\widetilde{n}_k| \rfloor, & m_k &= \lfloor \widetilde{m}_k \rfloor, & & \\
b_j^{i,k} &\overset{\text{i.i.d.}}{\sim} \mathcal{N}\big(M_i, (M_i/10)^2\big), & \hat{b}_r^{i,k} &\overset{\text{i.i.d.}}{\sim} \mathcal{N}\big(M_i, (M_i/10)^2\big), & j &= 1, \ldots, n_k, \\
p_j^{i,k} &\overset{\text{i.i.d.}}{\sim} \mathcal{N}\big(M_i, (M_i/10)^2\big), & \hat{p}_r^{i,k} &\overset{\text{i.i.d.}}{\sim} \mathcal{N}\big(M_i/10, (M_i/10)^2\big), & r &= 1, \ldots, m_k. \\
d_j^{i,k} &= b_j^{i,k} + |p_j^{i,k}|, & \hat{d}_r^{i,k} &= \hat{b}_r^{i,k} + |\hat{p}_r^{i,k}|, & &
\end{aligned}
$$

The first subset $\{(b_j^{i,k}, d_j^{i,k})\}_{j=1}^{n_k}$ represents the informative points in the diagram, while $\{(\hat{b}_r^{i,k}, \hat{d}_r^{i,k})\}_{r=1}^{m_k}$ captures the noisy points near the diagonal. The parameter $N$ controls the expected cardinality of the informative part, whereas $N'$ governs the noisy part. Note that the variables $\{\widetilde{m}_k\}$ are sampled uniformly in $[1, N'-1]$, and therefore have much higher variance than $\{\widetilde{n}_k\}$.

In our experiments, we set $N = 50$, $M_1 = 100$, and $M_2 = 70$, and consider $N' \in \{10, 1000, 5000, 10000\}$.

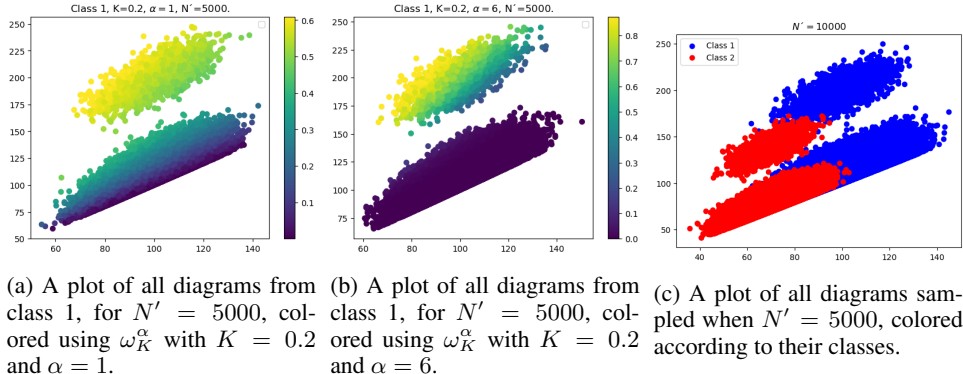

(a) A plot of all diagrams from class 1, for $N' = 5000$, colored using $\omega_K^\alpha$ with $K = 0.2$ and $\alpha = 1$.

(b) A plot of all diagrams from class 1, for $N' = 5000$, colored using $\omega_K^\alpha$ with $K = 0.2$ and $\alpha = 6$.

(c) A plot of all diagrams sampled when $N' = 5000$, colored according to their classes.

Figure 6: Figures related to Appendix B.1. We display the data and the effect of increasing $\alpha$ to sharpen the transition between low and high weight points. The parameter $K = 0.2$ was chosen by visual inspection to separate the noisy points from the others.

For each value of $N'$, we compute PSs from the sampled diagrams and then perform hierarchical clustering with average linkage on the resulting PS distance matrix. Cutting the dendrogram to obtain two clusters, whose partition we evaluate with the Rand index.

We report the results obtained with PS using $K = 0.2$ (chosen by visual inspection; see Figure 6) and two values of $\alpha$, namely $\alpha = 1$ and $\alpha = 6$. For reference, we also compare with PIs (with parameters selected on a grid to maximize clustering accuracy), using persistence as the weighting function, first with $q = 1$ and then with $q = 6$. The results in Table 2 indicate that higher levels of stability, corresponding to larger values of $q$ and $\alpha$, lead to improved clustering performance.

Table 2: Rand index for different methods and noise levels $N'$.

| Method | $N' = 10$ | $N' = 1000$ | $N' = 5000$ | $N' = 10000$ |
|---|---|---|---|---|
| PS $\alpha = 1$ | 0.97 | 0.66 | 0.59 | 0.60 |
| PS $\alpha = 6$ | 0.97 | 0.97 | 0.92 | 0.96 |
| PI $q = 1$ | 1.00 | 1.00 | 0.71 | 0.81 |
| PI $q = 6$ | 1.00 | 1.00 | 1.00 | 1.00 |

## B.2 THE ROLE OF $K$: AN UNSUPERVISED SIMULATION

In this section, we present a simulation study illustrating that the main effect of the parameter $K$ is to modulate the contribution of points near the diagonal, thereby altering the geometry of the vectorizations only when many noisy points are present. To demonstrate this, we consider a setting where points are sampled uniformly on a large bounded subset of $\{(x, y) \in \mathbb{R}^2 : x < y\}$, i.e. without an explicit concentration of points near the diagonal, and examine how varying $K$ affects pairwise relationships between diagrams. We extend this experiment to all vectorization methods considered in the paper, assessing how well each topological summary reflects the Wasserstein geometry of the underlying persistence diagrams.

To this end, we generate independent random pairs of persistence diagrams and compare their 1-Wasserstein distance with the distances induced by the corresponding vectorizations (or the sliced Wasserstein distance in the case of SWK). Specifically, we sample $500$ independent pairs of diagrams, compute their 1-Wasserstein distances, and correlate these values with the distances between their vectorized representations.

Each diagram is generated as follows: we first draw an integer $N$ uniformly from $[1, 10^4]$, then sample an $N \times 2$ matrix with entries independently and uniformly drawn from $[1, 10^4]$. Finally, we add the first coordinate to the second to enforce the constraint $x < y$.

Table 3: Correlation with 1-Wasserstein distance for different parameter settings (values rounded to $10^{-4}$).

| SW $(s)$ | PS $(K)$ | PI $\big((N_{\mathrm{pix}}, N_\sigma)\big)$ | PL | PSpl $\big((h, \mathrm{iter})\big)$ |
|---|---|---|---|---|
| $5 : 0.9989$ | $0 : 0.9993$ | $(50, 0.001) : 0.9283$ | $0.9650$ | $(10, 5) : 0.1831$ |
| $10 : 0.9989$ | $0.001 : 0.9993$ | $(50, 0.1) : 0.9220$ | – | $(10, 10) : 0.1752$ |
| $20 : 0.9989$ | $0.01 : 0.9993$ | $(50, 1) : 0.9216$ | – | $(20, 5) : 0.2609$ |
| – | $0.1 : 0.9994$ | $(50, 10) : 0.9214$ | – | $(20, 10) : 0.2433$ |
| – | $0.5 : 0.9996$ | $(100, 0.001) : 0.9283$ | – | $(40, 5) : 0.3805$ |
| – | – | $(100, 0.1) : 0.9220$ | – | $(40, 10) : 0.3316$ |
| – | – | $(100, 1) : 0.9216$ | – | – |
| – | – | $(100, 10) : 0.9214$ | – | – |

The results, reported in Table 3 for different parameter choices across methods, show the impact of these hyperparameters in an unsupervised setting. The ranges explored for PI and PSpl are chosen based on the best-performing configurations in the case studies, while for PS we fix $N_\theta = 100$ and $\alpha = 1$.

Table 3 also provides a practical validation of Corollary 1: for this generative process, PS achieves the highest correlation with the original Wasserstein distances, independently of the choice of $K$. The next best performance, with almost identical results, is obtained by the sliced Wasserstein distance, which is the other vectorization method for which a form of inverse continuity has been established Carriere et al. (2017). By contrast, PSpl exhibits by far the weakest agreement with the Wasserstein geometry, suggesting that it may be ill-suited for unsupervised analyses, as confirmed by Table 1.

## C  Computational Aspects, Runtimes, and Additional Implementation Details

The computation of a PS scales linearly with both the number of points in the diagram and the size of the evaluation grid. As shown in Equation (3), it reduces to evaluating standard mathematical functions in one or two variables. Since these evaluations are independent across points, the process can be efficiently parallelized. As a result, PSs are potentially cheaper than PIs, which require binning and integration, and PLs, whose fastest known algorithm has complexity $O(n \log n + nN)$ (Bubenik & Dłotko, 2017), where $n = \#D$ and $N$ is the number of nonzero landscapes. Approximating SWK (and treating the number of slices $s$ as a constant) incurs a computational cost of $O(n \log n)$ (Carriere et al., 2017). When evaluated on a grid, PSs have the same dimensionality as PIs on a comparable grid, since both are scalar fields on 2D manifolds.

For PSpls we ported in `python` the matlab code found in `https://github.com/ZC119/PB`, since the grid size of PSpls was never exceeding $50^2$, even without optimizing the code, we were able to run all the needed experiments. Still, since our code is not optimized, the upcoming runtimes comparison will not feature PSpls.

### C.1  Runtime Simulation

In this simulation, we aim to illustrate the linear computational cost of PSs. To this end, we randomly sample PDs of varying sizes and compute the corresponding PS, PL, and PI representations. To keep the comparison fair, we choose grids so that the resulting vectorizations have comparable sizes: PSs are computed on a grid of shape $(100, 200)$ (so their spherical harmonics representation has 1250 coefficients), PIs on a grid of shape $(200, 100)$, and PLs on a grid of 500 points, yielding a representation whose size depends on the number of landscapes. We also report runtimes for PSs including the decomposition into spherical harmonics, so as to compare against the runtimes of the actually employed vectorization.

The generative process for the random PDs is straightforward: for each $N \in \{5, 10, 10^2, 10^3, 10^4, 2 \cdot 10^4\}$, we sample a matrix of shape $(N, 2)$ with entries drawn independently and uniformly from

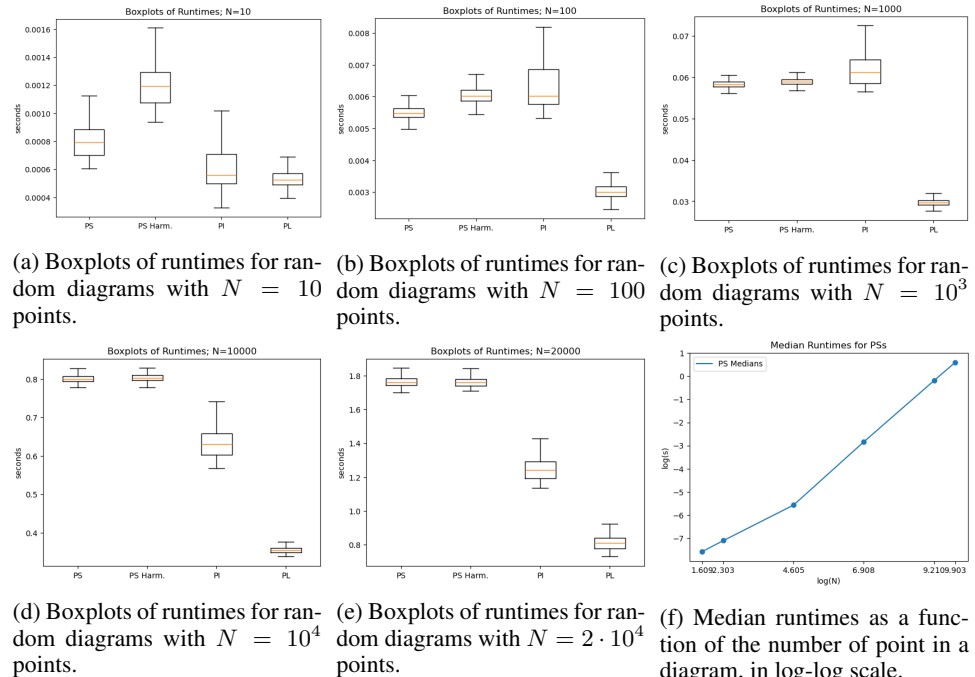

(a) Boxplots of runtimes for random diagrams with $N = 10$ points.

(b) Boxplots of runtimes for random diagrams with $N = 100$ points.

(c) Boxplots of runtimes for random diagrams with $N = 10^3$ points.

(d) Boxplots of runtimes for random diagrams with $N = 10^4$ points.

(e) Boxplots of runtimes for random diagrams with $N = 2 \cdot 10^4$ points.

(f) Median runtimes as a function of the number of point in a diagram, in log-log scale.

Figure 7: Figures associated with Appendix C. The boxplots indicate that PSs achieve runtimes comparable to established implementations of PIs and PLs, while Figure 7f illustrates an almost perfectly linear growth of the median PS runtime with respect to the number of points in a diagram.

$[1, 10^4]$. We then add the first coordinate to the second to enforce $x < y$. For each value of $N$, we generate 1000 random diagrams.

The results, reported in Figure 7, clearly display a linear relationship between runtime and the number of points in the diagrams (see in particular Figure 7f), and show that the spherical-harmonics-based vectorization is very efficient. At the same time, Figure 7 indicates that PIs and PLs are often computed more quickly. Recall that, for PIs and PLs, we relied on the highly optimized `scikit-tda persim` module. We therefore expect that our current PS implementation could be further optimized and potentially integrated into this module.

## D  PERSLAY ARCHITECTURES

In our experiments with PersLay, each persistence diagram is first rescaled to lie in $[0, 1] \times [0, 2]$ and then processed by a multi-branch architecture combining up to four topological channels with a small dense head. Between $N_{\text{Gauss}} = 1$ and $N_{\text{Gauss}} = 3$ branches employ `GaussianPerslayPhi` at different image resolutions, namely $10 \times 10$, $20 \times 20$, or $50 \times 50$, on the fixed birth–death box $[0, 1] \times [0, 2]$; the corresponding variance parameter is learned during training. The remaining branch uses `TentPerslayPhi` evaluated on a grid of $N_{\text{tent}}$ samples in $[0, 2]$, with $N_{\text{tent}}$ ranging from 100 to 1000, and the sample locations treated as trainable parameters. In all branches we adopt `GaussianMixturePerslayWeight` with a mixture of $K_{\text{Gauss}}$ components on the birth–death plane, where $K_{\text{Gauss}}$ varies between 5 and 15. Within each branch, weighted features are aggregated over points by sum pooling and passed through a branch-specific batch-normalization layer; the Gaussian-image outputs and the tent output are then flattened and concatenated into a single feature vector.

On top of this representation we place a small fully connected head: a first dense layer with $N_{\text{ReLU}}$ ReLU units, where $N_{\text{ReLU}} \in \{16, \ldots, 48\}$, followed by a dropout layer with rate $r_{\text{drop}} \in [0, 0.4]$, and a second dense layer with $N_{\text{ReLU}}/2$ ReLU units. This is followed by a task-specific output layer: for classification we use a softmax layer with $C$ (number of classes) units and optimize categorical cross-entropy, while for regression we use a single linear unit optimized with mean squared error.

| Dataset | $N_{\text{Gauss}}$ | $N_{\text{tent}}$ | $K_{\text{Gauss}}$ | $N_{\text{ReLU}}$ | $r_{\text{drop}}$ |
|---|---|---|---|---|---|
| Eyeglasses | 1 | 0 | 10 | 16 | 0 |
| Tecator | 3 | 500 | 15 | 16 | 0.15 |
| Growth | 3 | 500 | 15 | 16 | 0.25 |
| NOx | 3 | 500 | 15 | 16 | 0.25 |
| DYN_SYS | 3 | 500 | 15 | 48 | 0.40 |
| ENZYMES_JACC | 3 | 500 | 15 | 48 | 0.40 |
| POWER | 3 | 500 | 15 | 48 | 0.40 |
| SHREC14 | 3 | 500 | 15 | 48 | 0.40 |

Table 4: PersLay hyperparameters used for each case study.

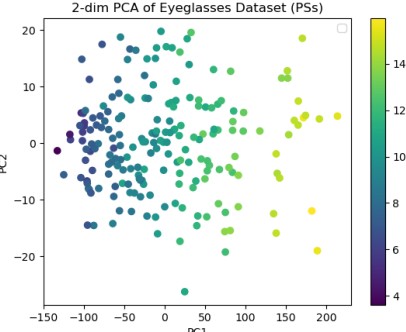

Figure 8: PS-based PCA of the Eyeglasses dataset.

In all cases, the network is trained with the Adam optimizer. During our analyses, we explored different configurations within the parameter ranges described above, monitoring training runs of 20 to 50 epochs (depending on network size) with batch size 32, and then selected the final architectures reported in Table 4. All other experimental choices (e.g. train–test splits) were kept consistent with the remaining pipelines.

As already mentioned in the main text, we were unable to identify a PersLay architecture that achieved competitive performance on the Eyeglasses dataset, even after adjusting the train–test ratio to increase the amount of training data. We therefore focused on very shallow networks, motivated by the low intrinsic dimensionality of this case study: PCA on the spherical harmonics representation of PS shows that the first principal component alone explains 0.985 of the variance (see Figure 8). In the end, the best-performing architecture used a single `GaussianPerslayPhi` layer with resolution $10 \times 10$, no `TentPerslayPhi` branch, no dropout, and a Gaussian mixture with $K_{\text{Gauss}} = 10$ components.

Lastly, we note that, by choosing as *point transformation* (Carrière et al., 2020)

$$p \longmapsto \text{ReLU}\big(\langle v, (1, p)\rangle\big), \qquad v \in \mathbb{S}^2,$$

one essentially recovers the basic building block of PSs. This suggests that PSs could, in principle, be integrated directly into a PersLay architecture as an additional topological channel, for instance by treating a grid on $\mathbb{S}^2$ as a set of trainable features, along with the weights, in close analogy with how the landscape-based transformation is handled.

# E    ABLATION STUDIES

Lastly, we present a set of ablation studies designed to isolate the practical impact of individual PS parameters on performance. To this end, we considered the four datasets from Bandiziol & De Marchi (2024) and, in each experiment, varied a single parameter while keeping all others fixed. These datasets were chosen because, for a fixed method, they exhibit substantial variability in classification accuracy: some case studies are considerably more challenging than others.

The default parameter configuration, i.e., the values used whenever a parameter was not under ablation, was set to $2N_\theta = 40$, $K = 10^{-2}$, and $\alpha = 1$. The number of estimators in the random forest was fixed to 100 throughout.

We first varied $2N_\theta \in \{30, 40, 50, 60\}$. We then explored $K \in \{-1, 0, 0.01, 0.1, 0.5\}$, where $K = -1$, $K = 0$ respectively denote, with an abuse of notation, the weighting function $p \mapsto (y - x)/2$ (used by Gotovac Dogaš & Mandarić (2025)), and the constant weighting function equal to 1. Finally, we examined $\alpha \in \{1, 2, 4, 8\}$.

The results, reported in Table 5, show that PSs are very robust to parameter choices in supervised case studies. This is largely because, in supervised settings, the learning algorithm can compensate for suboptimal weighting functions and still effectively suppress noise in the data (which is precisely the role of $K$ and $\alpha$). In unsupervised situations, the choice should be guided by Appendix B.

Table 5: Ablation Studies. Mean $\pm$ standard deviation (over 10 runs) for each dataset and parameter setting.

|  | POWER | DYN_SYS | SHREC14 | ENZYMES_JACC |
|---|---|---|---|---|
| $2N_\theta = 30$ | $0.760 \pm 0.026$ | $0.779 \pm 0.035$ | $0.909 \pm 0.020$ | $0.364 \pm 0.028$ |
| $2N_\theta = 40$ | $0.764 \pm 0.019$ | $0.783 \pm 0.018$ | $0.887 \pm 0.035$ | $0.354 \pm 0.028$ |
| $2N_\theta = 50$ | $0.768 \pm 0.016$ | $0.817 \pm 0.021$ | $0.881 \pm 0.037$ | $0.378 \pm 0.022$ |
| $2N_\theta = 60$ | $0.754 \pm 0.015$ | $0.809 \pm 0.026$ | $0.893 \pm 0.032$ | $0.386 \pm 0.026$ |
| $K = -1$ | $0.481 \pm 0.012$ | $0.155 \pm 0.012$ | $0.031 \pm 0.011$ | $0.134 \pm 0.010$ |
| $K = 0$ | $0.763 \pm 0.013$ | $0.809 \pm 0.029$ | $0.847 \pm 0.023$ | $0.349 \pm 0.027$ |
| $K = 0.01$ | $0.773 \pm 0.018$ | $0.790 \pm 0.029$ | $0.884 \pm 0.023$ | $0.362 \pm 0.041$ |
| $K = 0.1$ | $0.766 \pm 0.024$ | $0.806 \pm 0.022$ | $0.912 \pm 0.031$ | $0.371 \pm 0.027$ |
| $K = 0.5$ | $0.763 \pm 0.017$ | $0.801 \pm 0.015$ | $0.907 \pm 0.033$ | $0.357 \pm 0.025$ |
| $\alpha = 1$ | $0.770 \pm 0.027$ | $0.783 \pm 0.037$ | $0.893 \pm 0.028$ | $0.364 \pm 0.031$ |
| $\alpha = 2$ | $0.769 \pm 0.020$ | $0.805 \pm 0.030$ | $0.877 \pm 0.026$ | $0.368 \pm 0.035$ |
| $\alpha = 4$ | $0.755 \pm 0.019$ | $0.799 \pm 0.017$ | $0.886 \pm 0.030$ | $0.347 \pm 0.038$ |
| $\alpha = 8$ | $0.774 \pm 0.015$ | $0.803 \pm 0.022$ | $0.908 \pm 0.028$ | $0.371 \pm 0.024$ |

Remarkably, the only choice which yielded considerably worse results is $K = -1$, which we used to indicate the weighting function used in Gotovac Dogaš & Mandarić (2025).

Performance is also highly stable with respect to the choice of grid (and thus the dimension of the vectorization), despite the different sizes and difficulty levels of the considered case studies. We attribute this to the fact that each PS is a Lipschitz function on the sphere, and therefore well-behaved and not overly difficult to approximate via spherical harmonics expansions.

As noted in Section 4.3, the McGill 3D Shapes dataset is the only exception to this pattern: despite its modest size, substantially increasing the dimensionality of the PS vectorization led to clearly improved results.

## F  PROOFS OF THE RESULTS

**Proposition 4.** *Set* $\lambda(p) := \frac{y-x}{2\|(1,p)\|_2}$. *The following are stable weightings:*

$$\widetilde{\omega}(p) = \lambda(p)^\alpha, \qquad \omega_K^\alpha(p) = \frac{2}{\pi} \arctan\left(\frac{\lambda(p)^\alpha}{K^\alpha}\right),$$

*for any* $K > 0$ *and* $\alpha \geq 1$. *They are also effective weightings for* $\alpha = 1$.

*Proof.* The functions $\Gamma_\omega$ have the following forms:

$$\Gamma_{\widetilde{\omega}}(x,y) = \frac{(y-x)^\alpha}{2^\alpha \|(1,x,y)\|_2^\alpha}(1,x,y);$$

$$\Gamma_{\omega_K}(x,y) = \frac{2}{\pi} \arctan\left(\frac{(y-x)^\alpha}{2^\alpha K^\alpha \|(1,x,y)\|_2^\alpha}\right)(1,x,y).$$

Lipschitzianity is obtained because the components of the functions $\Gamma_{\widetilde{\omega}}$ and $\Gamma_{\omega_K}$ are differentiable and have bounded partial derivatives on $\mathbb{R}^2_{x<y}$.

To check the norm condition for stability, we write down the expressions of $\|\Gamma_\omega\|_2$:

$$\|\Gamma_{\widetilde{\omega}}(x,y)\|_2 = \frac{(y-x)^\alpha}{2^\alpha \|(1,x,y)\|_2^{\alpha-1}};$$

$$\|\Gamma_{\omega_K}(x,y)\|_2 = \frac{2}{\pi} \arctan\left(\frac{(y-x)^\alpha}{2^\alpha K^\alpha \|(1,x,y)\|_2^\alpha}\right)\|(1,x,y)\|_2.$$

At this point, we observe that:

$$\frac{(y-x)^{\alpha-1}}{2^{\alpha-1}\|(1,x,y)\|_2^{\alpha-1}} \in [0,1);$$

and that:

$$\arctan\left(\frac{(y-x)^{\alpha}}{2^{\alpha}K^{\alpha}\|(1,x,y)\|_2^{\alpha}}\right)\|(1,x,y)\|_2 \le \frac{(y-x)^{\alpha}}{2^{\alpha}K^{\alpha}\|(1,x,y)\|_2^{\alpha-1}}.$$

The first observation is enough to prove stability for $\widetilde{\omega}$, while the second and the first observations, combined, prove it for $\omega_K$.

Now we prove that both weightings are effective for $\alpha = 1$, exploiting Equation (2). Note that the functions have become:

$$\|\Gamma_{\widetilde{\omega}}(x,y)\|_2 = \frac{(y-x)}{2} = \|p - \Delta\|_\infty;$$

$$\|\Gamma_{\omega_K}(x,y)\|_2 = \frac{2}{\pi}\arctan\left(\frac{(y-x)}{2K\|(1,x,y)\|_2}\right)\|(1,x,y)\|_2.$$

To see that $\Gamma_{\widetilde{\omega}}$ is effective, it suffices to observe that, plugging the expression of $\|\Gamma_{\widetilde{\omega}}\|_2$ in Equation (2), we directly obtain the thesis.

Now we deal with $\Gamma_{\omega_K}$. Set $\mu_{D_n} = \sum_{p \in D_n} a_{n,p}\delta_p$.

We rewrite Equation (2) as:

$$\lim_{r \to \infty}\sup_n \sum_{p \in D_n, \|p\|_2 > r}\|\Gamma_{\omega_K}(p)\|_2 \le \lim_{r \to \infty}\sup_n \sum_{p \in D_n, \|p\|_2 > r} a_{n,p}\|\Gamma_{\omega_K}(p)\|_2 \to 0.$$

Thus, for every $\varepsilon > 0$, there is $R > 0$ such that, for every $r > R$ the following holds:

$$\sup_n \sum_{p \in D_n, \|p\|_2 > r}\|\Gamma_{\omega_K}(p)\|_2 < \varepsilon. \tag{6}$$

Hence, for every $n$, we have:

$$\sum_{p=(x,y)\in D_n, \|p\|_2 > r}\arctan\left(\frac{(y-x)}{2K\|(1,x,y)\|_2}\right) < \frac{\pi\varepsilon}{2r}. \tag{7}$$

In particular, Equation (7) implies $\frac{(y-x)}{2K\|(1,x,y)\|_2} \to 0$ for $r \to \infty$. Equivalently, for every $C > 0$, there is $r_C$ such that $\frac{(y-x)}{2K\|(1,x,y)\|_2} < C$.

The key observation now, is that, due to the concavity of $z \mapsto \arctan(z)$, which is easy to see due to the strict monotonicity of its derivative $\frac{1}{1+z^2}$, we have:

$$\frac{\arctan(\varepsilon)}{\varepsilon}z \le \arctan(z) \le z \tag{8}$$

for every $z \in [0,\varepsilon]$, and every fixed $\varepsilon \ge 0$. In fact, $z \mapsto \frac{\arctan(\varepsilon)}{\varepsilon}z$ is the straight line joining $(0,0)$ and $(\varepsilon, \arctan(\varepsilon))$. Thus, for every $C > 0$ and for every $r_C$ such that $\frac{(y-x)}{2K\|(1,x,y)\|_2} < C$, we have for every $n$:

$$C'\sum_{p \in D_n, \|p\|_2 > r_C}\frac{a_{n,p}(y-x)}{2} \le \sum_{p \in D_n, \|p\|_2 > r_C}\frac{2a_{n,p}}{\pi}\arctan\left(\frac{(y-x)}{2K\|(1,x,y)\|_2}\right)\|(1,x,y)\|_2,$$

with $C' = \frac{2\arctan(C)}{\pi CK}$.

In particular, we can find $C' > 0$ such that:

$$C'\lim_{r \to \infty}\sup_n \text{Pers}_{B_r^c}(\mu_{D_n}) \le \lim_{r \to \infty}\sup_n \sum_{p \in D_n, \|p\|_2 > r} a_{n,p}\|\Gamma_{\omega_K}(p)\|_2 \to 0.$$

Which proves the thesis. $\qquad\square$

**Theorem 4.** *Let $\mu_D, \mu'_D$ be PDs and let $\omega : \mathbb{R}^2 \to (0, 1]$ be a stable weighting. We have:*

$$d_H(Z_{\mu_D^\omega}, Z_{\mu_{D'}^\omega}) \leq \sqrt{2} \max\{C, C'\} W_1(\mu_D, \mu_{D'}),$$

*with $C, C' > 0$ being the stability constants of $\omega$.*

*Proof.* Consider $\mu_D = \sum_{p \in D} a_p \delta_p, \mu_{D'} = \sum_{q \in D'} b_q \delta_q$ and a partial matching $\gamma$ between them. Without loss of generality, suppose $C = \max\{C, C'\}$.

A generic point in $Z_{\mu_D^\omega}$ has the form:

$$P = \sum_{p \in D} a_p s_p \Gamma_\omega(p) \in Z_{\mu_D^\omega},$$

with $s_p \in [0, 1]$. We start by considering $p \in D_\gamma$ and $P \in Z_{\mu_D^\omega}$ with the following form:

$$P = \gamma_p \Gamma_\omega(p). \tag{9}$$

Note that, by definition, $\gamma_p \in \mathbb{N}$ and $\gamma_p \geq 1$.

Consider the point:
$$Q = \gamma_p \Gamma_\omega(\gamma(p)).$$

Since $\gamma_p \leq b_{\gamma(p)}$, there is $s \in [0, 1]$ such that $sb_{\gamma(p)} = \gamma_p$. Thus, $Q \in Z_{\mu_{D'}^\omega}$.

We have:

$$\|P - Q\|_2 \leq \gamma_p \|\Gamma_\omega(p) - \Gamma_\omega(\gamma(p))\|_2 \leq \gamma_p C \|p - \gamma(p)\|_2.$$

For $P$ in the form of Equation (9), we define $\Phi(P) := Q$.

Consider now a generic $P \in Z_{\mu_D^\omega}$:

$$P = \sum_{p \in D} a_p s_p \Gamma_\omega(p) = \sum_{p \in D_\gamma} \gamma_p s_p \Gamma_\omega(p) + \sum_{p \in D_\gamma} (a_p - \gamma_p) s_p \Gamma_\omega(p) + \sum_{p \in D - D_\gamma} a_p s_p \Gamma_\omega(p),$$

We build $Q$ as follows:

$$Q = \sum_{p \in D_\gamma} s_p \Phi(\gamma_p \Gamma_\omega(p)).$$

We have:

$$\|P - Q\|_2 \leq \sum_{p \in D_\gamma} s_p \gamma_p \|\Gamma_\omega(p) - \Gamma_\omega(\gamma(p))\|_2 + \sum_{p \in D_\gamma} s_p (a_p - \gamma_p) \|\Gamma_\omega(p)\|_2 + \\ \sum_{p \in D - D_\gamma} s_p a_p \|\Gamma_\omega(p)\|_2. \tag{10}$$

Plugging into Equation (10) the following facts:

1. $\|\Gamma_\omega(p)\|_2 \leq C \|p - \Delta\|_\infty$;

2. $\|\cdot\|_2 \leq \sqrt{2} \|\cdot\|_\infty$;

3. $s_p \leq 1$ for every $p \in D$,

we obtain:

$$\|P - Q\|_2 \leq$$

$$C \left( \sqrt{2} \sum_{p \in D_\gamma} \gamma_p \|p - \gamma(p)\|_\infty + \sum_{p \in D_\gamma} (a_p - \gamma_p) \|p - \Delta\|_\infty + \sum_{p \in D - D_\gamma} a_p \|p - \Delta\|_\infty \right).$$

Since we can do this construction for any partial matching $\gamma$, for every $P \in Z_{\mu_D^\omega}$, we found $Q$ such that:

$$\|P - Q\|_2 \leq \sqrt{2} \, C \, W_1(\mu_D, \mu_{D'}).$$

Reversing the role of $\mu_D$ and $\mu_{D'}$ we obtain the thesis. $\qquad\square$

**Theorem 5.** *Let $\{\mu_{D_n}\}_{n \in \mathbb{N}}$ be a sequence of PDs such that $d_H(Z_{\mu_{D_n}^\omega}, Z_{\mu_D^\omega}) \to 0$, with $\omega : \mathbb{R}^2 \to (0, 1]$ being an effective weighting. Then $W_1(\mu_{D_n}, \mu_D) \to 0$.*

*Proof.* Combining Proposition 2 with the assumption $d_H(Z_{\mu_{D_n}^\omega}, Z_{\mu_D^\omega}) \to 0$, we obtain that $\mu_{D_n}^\omega \xrightarrow{w} \mu_D^\omega$ and that $\{\mu_{D_n}^\omega\}_{n \in \mathbb{N}}$ is uniformly integrable.

Let $f \in C_c(\mathbb{R}^2_{x<y})$ and set $E := \mathrm{supp}(f) \subset \mathbb{R}^2_{x<y}$. Since $E$ is compact and $E \cap \Delta = \varnothing$, by continuity and strict positivity of $\omega$ we have $m_E := \min_{p \in E} \omega(p) > 0$. Hence $h := f/\omega$ is continuous on $E$ and bounded by $\|f\|_\infty / m_E$. Extending $h$ by $0$ outside $E$ (note that $f$ is $0$ on $\partial E$), we obtain a bounded continuous function on $\mathbb{R}^2$ and therefore, by weak convergence,

$$\langle \mu_{D_n}, f \rangle = \int f \, d\mu_{D_n} = \int \omega \frac{f}{\omega} \, d\mu_{D_n} = \int h \, d\mu_{D_n}^\omega = \langle \mu_{D_n}^\omega, h \rangle \longrightarrow \langle \mu_D^\omega, h \rangle = \langle \mu_D, f \rangle.$$

Thus $\mu_{D_n} \xrightarrow{v} \mu_D$. If we prove $\mathrm{Pers}(\mu_{D_n}) \to \mathrm{Pers}(\mu_D)$, then we can conclude by Theorem 1.

By uniform integrability we have:

$$\lim_{r \to \infty} \sup_{n \in \mathbb{N}} \int_{B_r^c} \omega(p) \| p \|_2 \, d\mu_{D_n}(p) = \lim_{r \to \infty} \sup_{n \in \mathbb{N}} \int_{B_r^c} \| p \|_2 \, d\mu_{D_n}^\omega(p) = 0.$$

Since $\omega$ is an effective weighting, this implies:

$$\lim_{r \to \infty} \sup_{n \in \mathbb{N}} \mathrm{Pers}_{B_r^c}(\mu_{D_n}) \to 0,$$

For any $r \geq 0$ we can write:

$$2 \, \mathrm{Pers}(\mu_{D_n}) = \int_{\mathbb{R}^2_{x<y}} (y - x) d\mu_{D_n}((x,y)) = \int_{B_r^c} (y - x) d\mu_{D_n}((x,y)) + \int_{B_r} (y - x) d\mu_{D_n}((x,y)).$$

Similarly, we can write:

$$2 \, \mathrm{Pers}(\mu_D) = \int_{\mathbb{R}^2_{x<y}} (y - x) d\mu_D((x,y)) = \int_{B_r^c} (y - x) d\mu_D((x,y)) + \int_{B_r} (y - x) d\mu_D((x,y)).$$

If $r$ is big enough, being $D$ finite, we have $\mathrm{supp}(D) \subset B_r$, and so $\int_{B_r^c} (y - x) d\mu_D((x,y)) = 0$ and $\int_{B_r} (y - x) d\mu_D((x,y)) = 2 \, \mathrm{Pers}(\mu_D)$.

Fix some $r$ big enough so that the above holds. Since $B_r$ is compact we can write a positive test function $g : \mathbb{R}^2_{x<y} \to [0, 1]$ such that:

- $g$ is continuous;
- $g \equiv 1$ on $B_r$;
- $\mathrm{supp}(g)$ is compact.

For such a $g$, we obtain:

$$0 \leq \int_{B_r} (y - x) d\mu_{D_n}((x,y)) \leq \int_{\mathbb{R}^2_{x<y}} g(x,y)(y - x) d\mu_{D_n}((x,y)).$$

Moreover, using vague convergence, we get:

$$\int_{\mathbb{R}^2_{x<y}} g(x,y)(y - x) d\mu_{D_n}((x,y)) \xrightarrow{n} 2 \, \mathrm{Pers}(\mu_D).$$

But:

$$\int_{\mathbb{R}^2_{x<y}} g(x,y)(y-x)d\mu_{D_n}((x,y)) =$$

$$\int_{B_r} g(x,y)(y-x)d\mu_{D_n}((x,y)) + \int_{B_r^c} g(x,y)(y-x)d\mu_{D_n}((x,y)) =$$

$$\int_{B_r} (y-x)d\mu_{D_n}((x,y)) + \int_{B_r^c} g(x,y)(y-x)d\mu_{D_n}((x,y)).$$

Thus:

$$0 \le \int_{\mathbb{R}^2_{x<y}} g(x,y)(y-x)d\mu_{D_n}((x,y)) - \int_{B_r} (y-x)d\mu_{D_n}((x,y)) =$$

$$\int_{B_r^c} g(x,y)(y-x)d\mu_{D_n}((x,y)) \le \int_{B_r^c} (y-x)d\mu_{D_n}((x,y)).$$

Putting the pieces together, for every $\varepsilon > 0$ there exist $r_\varepsilon$ and $N_\varepsilon$ such that, for every $n \ge N_\varepsilon$:

$$\mid 2\operatorname{Pers}(\mu_{D_n}) - \int_{B_{r_\varepsilon}} (y-x)d\mu_{D_n}((x,y)) \mid \le \varepsilon,$$

$$\mid 2\operatorname{Pers}(\mu_D) - \int_{\mathbb{R}^2_{x<y}} g(x,y)(y-x)d\mu_{D_n}((x,y)) \mid \le \varepsilon,$$

$$0 \le \int_{\mathbb{R}^2_{x<y}} g(x,y)(y-x)d\mu_{D_n}((x,y)) - \int_{B_{r_\varepsilon}} (y-x)d\mu_{D_n}((x,y)) \le \varepsilon,$$

entailing $2 \mid \operatorname{Pers}(\mu_{D_n}) - \operatorname{Pers}(\mu_D) \mid \le 3\varepsilon$, concluding the proof. $\square$

