# OpenReview forum: "PERSISTENCE SPHERES: BI-CONTINUOUS REPRESENTATIONS OF PERSISTENCE DIAGRAMS."
_ICLR.cc/2026/Conference — ICLR 2026 Poster_

### Official Review · Reviewer_YUj9 · 2025-10-21

**Soundness:** 2
**Presentation:** 1
**Contribution:** 2
**Rating:** 2
**Confidence:** 5

**Summary:**

This paper proposes a bi-continuous representation (with respect to the 1-Wasserstein distance) of Persistence Diagrams (PDs) via the restricted support of lift zonoids, i.e. Persistence Sphere (PS). The property of being bi-continuous is theoretically proved. The experiments focus on regression and classification tasks.

**Strengths:**

Strengths:

- The bi-continuity w.r.t. Wasserstein distance is very interesting and important for ensuring PD correspondence, the key component in Topological Machine Learning.
- The background definitions and proofs are through and self-contained.
- The bi-continuity holds for different weight functions ws. This provides flexibility for the representation.
- The theoretical proofs for continuity theorems are sound.
- The datasets in experiments are extensive and the results are convincing.

**Weaknesses:**

On the presentation aspect:
- The discretization method of the Persistence Sphere, a restricted support function of lift zonoids is unclear.
- The paper focuses too much on the math notations. Only one visualization (Fig c) on the proposed method is provided. The lack of visual examples in $\mathbb{R}^3$ of the key components such as lift zonoid of a PD (definition 10) and PS (definition 11) makes these concepts difficult to understand intuitively and reduces readability.
- The background of Persistence Diagram, a multiset on the open half plane, is very limited. Maybe a detailed one including the filtration process could be included in the Appendix.

On the experiments aspect:
- The baselines (PI, PL, SWK) are very limited. More recent methods such as Persistence B-spline grids [1], an unsupervised method, should be included.
- The baselines are all unsupervised vectorization methods. Since the paper focus on supervised task like classification, I think it is necessary to compare PS with supervised vectorization like PersLay [2] to show whether PS can outperform supervised baselines.
- The paper gives computational analysis without showing the actual time cost of PS. A scale-up test would make the analysis more convincing.
- Although it is nice that the bi-continuity holds for different weight functions $w$s, when it comes to experiments, it would be necessary to show how the choice of weight function affects the results. An ablation study on the choice of weight function would be nice.

[1] Dong Z, Lin H, Zhou C, et al. Persistence B-spline grids: stable vector representation of persistence diagrams based on data fitting[J]. Machine Learning, 2024, 113(3): 1373-1420.

[2] Carrière M, Chazal F, Ike Y, et al. Perslay: A neural network layer for persistence diagrams and new graph topological signatures[C]//International Conference on Artificial Intelligence and Statistics. PMLR, 2020: 2786-2796.

**Questions:**

See Weaknesses.

---

> ### Author Response · Authors · 2025-11-12
>
> We thank the reviewer for the feedback and for suggesting an interesting reference that we were not previously aware of.
>
> As for the other reviewers, we organize our response into two main sections: QUESTIONS / COMMENTS (points for which we would appreciate further clarification) and POINTS of CONSENSUS (points where we are aligned with the reviewer’s observations).
>
> QUESTIONS / COMMENTS
>
> 1) On the choice of benchmarks.
>
>     We believe the remark concerning the benchmarks being “very limited” is somewhat severe. We considered what have consistently emerged as the most effective kernel embeddings, along with the two most widely used topological summaries (to the best of our knowledge). Nevertheless, we greatly appreciate the reference to the Persistence Splines paper, which we were not aware of. We will make every effort to include a comparison with this method, although we note that the available implementation on GitHub partly relies on Matlab, which complicates integration into our fully Python-based pipeline.
>
>     On the other hand, we believe that a comparison with the PersLay architecture would not be appropriate for the present paper. PersLay is not a single vectorization method, but rather a flexible architecture that can incorporate and optimize multiple schemes. In fact, Persistence Spheres could be seamlessly integrated into the PersLay framework by choosing the point transformation (as defined in the PersLay paper)
>
>     p \mapsto \max\{0, \langle v, (1,p)\rangle\}, \quad v \in \mathbb{S}^2,
>
>     which embeds our representation directly within the PersLay architecture.
>
> 2) On runtime comparisons.
>
>     Consistently with our reply to Reviewer D5QX, we consider overall cross-validation runtimes to be of limited relevance in our context (see the detailed discussion in that response).
>
>     If desired, we could add a runtime comparison restricted to evaluating explicit summaries (PS, PL, and PI) on diagram populations of varying sizes and cardinalities.
>
> 3) “The discretization method of the Persistence Sphere, a restricted support function of lift zonoids, is unclear.”
>
>     We would appreciate it if the reviewer could reformulate this comment. We are not sure which aspect this refers to. We discretize the functions using a grid on spherical coordinates and the associated volume form. Is that what the reviewer is referring to?
>
>
> POINTS of CONSENSUS
>
> 1) “Although it is nice that the bi-continuity holds for different weight functions, when it comes to experiments, it would be necessary to show how the choice of weight function affects the results.”
>
>     We thank the reviewer for highlighting that the role of the weight function in our vectorization is insufficiently discussed. Since this function (and its associated parameter) has a clear interpretation, similar to the weighting in PIs, we will add a dedicated paragraph elaborating on this choice.
>
>     We anticipate that varying the weight function, provided it satisfies the properties defined in the paper, essentially corresponds to selecting a band along the diagonal that identifies points likely to represent noise, which are then smoothly down-weighted by a Lipschitz step function.
>
> 2) “The background of Persistence Diagram, a multiset on the open half-plane, is very limited. Maybe a detailed one including the filtration process could be included in the Appendix.”
>
>     We thank the reviewer for pointing this out. Reviewer 4NTj made a similar remark, and we will therefore add a paragraph in the appendix with additional details about TDA and PDs.
>
> 3) “The paper focuses too much on the math notations. Only one visualization (Fig. c) on the proposed method is provided. The lack of visual examples of key components such as the lift zonoid of a PD (Definition 10) and PS (Definition 11) makes these concepts difficult to understand intuitively and reduces readability.”
>
>     We thank the reviewer for this valuable suggestion. Reviewer 4NTj made similar comments, and we will therefore add a figure and a brief explanatory sentence to help the reader intuitively follow these definitions.

---

> > ### Comment · Reviewer_YUj9 · 2025-11-13
> > **Response to the authors**
> >
> > Thank you for a quick reponse. I now understand the discretization method.
> >
> > - On the choice of benchmarks: As shown in Table 1, compared with existing methods, Psph did not have significantly improved performance. Since it is a PD embedding used in supervised task, I still think that Psph should be compared with PersLay (for example, Gaussian point transform) or integrated into the PersLay. If the proposed method can not outperform PersLay in a supervised task, then what is the point of using Psph in practice? I think the unsupervised nature of Psph can actually has advantage in unsupervised tasks such as clustering point clouds or graph. An unsupervised task could be used to demonstrate the breadth of Psph that it can be used both for supervised and unsupervised cases. Besidies, the comparison with the lastest method Persistence B-spline grids is necessary.
> >
> > In addition to the choice of benchmarks, there are needs for scale-up test and hyperparameter sensitivity analysis. And I think a major revision is needed for improving readability. Thus, I think this paper is not ready for publication due to bad readability and lack of necessary experiments. I will maintain my original score.

---

> ### Author Response · Authors · 2025-11-13
>
> Dear Reviewer YUj9,
>
> You are, of course, free to assign whatever score you deem appropriate.
> Moreover, nobody asked you to change your score prior to any revision to the paper.
> That said, we still appreciate that you pointed us to the Persistence Splines reference, which we were not aware of.
>
> However, we believe it would be appropriate to reconsider your Confidence score. What you suggest (using PersLay with a Gaussian point transformation) effectively corresponds to employing the persistence surface as the underlying vectorization method. This, up to a binning step, is equivalent to Persistence Images, which are already included in our benchmark.
>
> Or perhaps you actually meant to employ a modified Gaussian transform and obtain the deep set–like formulation by Hofer et al.? If so, then you might want to phrase your comment more carefully.

---

> ### Comment · Reviewer_YUj9 · 2025-11-13
> **Response to the authors**
>
> Dear Authors,
>
> You are right about the equivalence to Persistence Images. I apology for my earlier expression that did not make things clear. By PersLay with a Gaussian point transformation, I mean the vectorization method with learnable parameters. In the case of Gaussian point transformation, these parameters are the Gaussian centers. This was used as a supervised vectorization method for point cloud classification task in [1]. Other supervised baselines would also be fine.
>
> My point is that it is necessary to compare Psph with a supervised vectorization method. This supervised baseline could be used as the difficulty of conducting this task with topology summary, i.e. Persistence Diagram. And it is perfectly OK if Psph achieves lower accuracy than this supervised baseline because Psph could also be used in unsupervised tasks.
>
> I think the idea of Psph is good and the theoretical contribution is sound, my score is based on the fact that current version does not have enough empirical evidence to demonstrate the effectivenss and efficiency of the proposed method. And the minor readability issue. I would change my rating if the requested experiments (more baseline, scale-up test or actual running time, hyperparameter sensitivity analysis, ablation study of the weight function) are provided and analyzed properly.
>
>
> [1] Nishikawa N, Ike Y, Yamanishi K. Adaptive topological feature via persistent homology: Filtration learning for point clouds[J]. Advances in Neural Information Processing Systems, 2023, 36: 9131-9143.

---

### Official Review · Reviewer_9Ldx · 2025-10-29

**Soundness:** 4
**Presentation:** 4
**Contribution:** 3
**Rating:** 6
**Confidence:** 4

**Summary:**

The paper proposes a new way of vectorizing persistence diagrams (PD). The idea of the vectorization procedure is the following: for each point $(b, d)$ of a PD with multiplicity $k$, consider a line segment from the origin to the point $(k, kb, kd)$. Take the Minkowski sum of all these segments, it is a convex polytope in $\mathbb{R}^3$. Now map the PD to the support function of this polytope (restricted onto the unit sphere). In fact, one has to apply weighting (points that are closer to the diagonal must contribute less). The paper establishes conditions for the weighting function that guarantee that the mapping PD -> support function is a Lipschitz embedding with contiuous inverse. The experimental part demonstrates the efficiency of this vectorization technique on various classification problems, by comparing it to many other known methods.
NB: the unweighted vectorization scheme was proposed in the literature, and the paper contains the appropriate references and attribution; the actual contribution of the paper is weighted case, its theoretical properties and experimental evaluation.

**Strengths:**

- The paper is rather well-written and very clear, being also practically self-contained.
- The properties that the embedding needs are certainly important, not just from a theoretical viewpoint, but also practically. Therefore the criteria on the weighting established in the paper are a solid contribution.
- The experiments cover various problems and look convincing.
- The proposed vectorization method is embarassingly parallel and can be expressed by a very simple formula.

**Weaknesses:**

- functions on a sphere are hard to work with, unlike, say, persistent images which are honest-to-goodness 2D numerical arrays. This fact is acknowledged in the paper.
- the paper could include the results for some of the experiments that the unweighted case paper made.

**Questions:**

Why not use spherical harmonics? I mean, to represent the vectorization instead of splines one could use the Fourier decomposition. Or has it been tried and the splines perform better?

---

> ### Author Response · Authors · 2025-11-12
>
> We thank the reviewer for the attention devoted to the paper and the feedback.
>
> 1) “Functions on a sphere are hard to work with.”
>
>     We thank the reviewer for highlighting this point. We fully agree that this is arguably the most delicate aspect of our method, as working on the sphere introduces challenges not present in flat domains.
>
>     Our current solution is to rely on functional PCA on the sphere implemented via spherical splines, which provides a straightforward yet computationally demanding approach. We are aware of this limitation and are actively exploring alternatives. In future work, we plan to develop a dedicated package that integrates persistence spheres with specialized statistical and computational tools for efficient data analysis on spherical domains.
>
> 2) “The paper could include the results for some of the experiments that the unweighted case paper made.”
>
>     We thank the reviewer for this valuable suggestion. We agree that it would be interesting to include comparisons with the unweighted scheme (although we would not strictly refer to it as “unweighted,” since points are still weighted by their persistence). We will aim to revise the case studies to include this comparison, where feasible within the page limits.
>
> 3) “Why not use spherical harmonics? I mean, to represent the vectorization instead of splines one could use the Fourier decomposition. Or has it been tried and the splines perform better?”
>
>     We greatly appreciate this suggestion. We had not considered spherical harmonics as an alternative representation, but we fully agree that they constitute a natural and elegant option. We will certainly explore this approach in future work, and evaluate its potential benefits relative to spline-based representations.

---

### Official Review · Reviewer_4NTj · 2025-10-30

**Soundness:** 3
**Presentation:** 3
**Contribution:** 3
**Rating:** 6
**Confidence:** 4

**Summary:**

This paper proposes a novel functional representation for Topological Data
Analysis, specifically for Persistence Diagrams (PDs).

The approach builds upon the theory connecting persistence diagrams with
measure theory and optimal transport. The functional associated with a
persistence diagram (viewed as a discrete measure) is then derived from the
support function of a zonoid (a convex set) constructed from this measure.

After the necessary background, the authors present the novel functionnal :
Persistence Sphere, and show that it satisfied multiple desirable properties
for a TDA representation, namely: stability, and separability.

The paper concludes with an experimental section, showcasing the performances
of this approach on various datasets.

**Strengths:**

- Well written.
 - Proofs seem to be sound
 -  The proposed construction sounds reasonable : stability (w.r.t.
 Wasserstein 1) is a must have for persistence diagram representations, and
 the separability property is very nice
 -  Computational complexity is linear
 - The code is available, and the code is reproducible in notebooks, with
 clear parameter selection.

**Weaknesses:**

- This paper is hard to read for someone who is not familiar with measure
 theory / TDA. For instance, there is no real intuition on what is a
 persistence diagram, how to interpret the matching cost, etc. As the paper
 presents a representation for PDs though, I think this approach can be argued
 to be fair.
 - The experiment section is weak. IIUC the SOTA is roughly from 2017 (sliced
 waserstein kernel).
 - l60-70 : Unless I misunderstood something, SWK is a stable and bicontinuous
 representations as well, with stronger separability properties (cf [SWK for
 PD, theorem 3.3]). The comparison to SWK is still on your favor though, since
 SWK is not an unsupervied vectorization method.
 - Def. 10 : a little picture/explanation could help
 - l193 : uniformly integrable is not def
 - l225 : "and p \in B_r^c"
 - l232 : link with uniform integrability?
 - l240 : "and α ≥ 1. They are also effective weightings for α = 1" what is
 the point of the last sentence ?
 - l244 : This is a bit fast here. I suppose the ReLU comes from the fact that
 the segments of the zonoid are based on 0.
 - l257 : the [Carrière Bauer] paper mentionned in the "Contribution"
 paragraph fits here ?
 - l273 : this is a bit fast as well here. I assume that $Z_\varnothing =
 \{0\}$?
 - l477-483 : some references should be added
 - l647 : K
 - l740 : I'm not sure the notation with \Phi help a lot
 - l777 : By prop. 2, [...] to conclude with thm 1
 - l787 : effective weighting
 - l793, 798: a factor 2 is missing

**Questions:**

- See weaknesses.
 - When comparing with the SWK, two question naturally arise : 1) is it
 possible to have a better separability bound ? e.g., f(W_1(...)) <= || PS...
 ||_p  for a non-trivial non-decreasing function f ? and 2) The sliced
 wasserstein distance can be exactly computed on small enough diagrams, since
 the combinatorix is finite, and can thus be decomposed into a finite number
 of cells. What about the || PS_1 - PS_2 ||_p of two diagrams ?

---

> ### Author Response · Authors · 2025-11-12
>
> We thank the reviewer for the constructive feedback, which was detailed and insightful, and pointed out several directions for improving the paper, including aspects of the proofs.
>
> As for the other reviewers, we split our rebuttal into two main sections: QUESTIONS / COMMENTS (points for which we would appreciate clarification) and POINTS of CONSENSUS (points where we are aligned with the reviewer’s remarks).
>
> QUESTIONS / COMMENTS
>
> 1) “l60–70: Unless I misunderstood something, SWK is a stable and bicontinuous representation as well, with stronger separability properties (cf. [SWK for PD, Theorem 3.3]). The comparison to SWK is still in your favor though, since SWK is not an unsupervised vectorization method.”}
>
>     We thank the reviewer for asking this specific question, which allows us to clarify this point. The statement is not entirely correct: the cited result does not prove that the sliced Wasserstein distance is metrically equivalent to the 1-Wasserstein distance, i.e., it is not a bi-Lipschitz result in general, but only holds for PDs with uniformly bounded cardinalities. A similar restriction appears in techniques developed by Mitra and Virk. Using the notation of the SWK paper, one can potentially have $M \to \infty$ without $d_1 \to 0$. This limitation does not occur for persistence spheres. We are still investigating whether a tighter Lipschitz bound can be obtained for persistence spheres when cardinalities are uniformly bounded, which is a topic for future research. The same applies for the separability bound mentioned in the questions.
>
> 2) “The experiment section is weak. IIUC the SOTA is roughly from 2017.”
>
>     We agree that this reference is from 2017, and also that PLs and PIs are not very recent. Nevertheless, even recent comparative studies, such as Bandiziol \& De Marchi, reinforce the relevance of these benchmarks. Reviewer YUj9 pointed out a more recent reference, which we were not aware of; we will attempt to integrate it into our comparison, despite some implementational barriers (Matlab vs Python), as explained in the reply to Reviewer YUj9.
>
> 3) “l193: uniformly integrable is not def.”
>
>     We thank the reviewer for the detailed reading. On this point, however, the definition of uniformly integrable sequences is given in Definition 4.
>
> 4) “l232: link with uniform integrability?”
>
>     This is well spotted. Indeed, this property interacts closely with uniform integrability, although the full extent of this interaction is contained in the proof of Theorem 2. It is not immediately clear how to make it more explicit without reducing conciseness and avoiding the technicalities which appear in the proof.
>
> 5) “l240: ‘and $\alpha \geq 1$. They are also effective weightings for $\alpha = 1$’ what is the point of the last sentence?”
>
>     The last sentence indicates that one can use $\alpha > 1$ and potentially optimize it for increased accuracy, but bi-continuity is guaranteed only for $\alpha = 1$. Likely it is an unnecessary remark: the term “effective” refers to Definition 13. We are also open to simplifying by always setting $\alpha = 1$.
>
> 6) “l740: I'm not sure the notation with $\Phi$ helps a lot.”
>
>     We are happy to modify the notation to improve clarity, but we would appreciate more precise feedback on which aspect of $\Phi$ is confusing.
>
> POINTS of CONSENSUS
>
> 1) “This paper is hard to read for someone who is not familiar with measure theory / TDA. For instance, there is no real intuition on what is a persistence diagram, how to interpret the matching cost, etc. As the paper presents a representation for PDs though, I think this approach can be argued to be fair.”
>
>     We thank the reviewer for pointing this out. It was noted also by Reviewer YUj9. We will add a paragraph in the appendix with additional details and intuition.
>
> 2) “Def. 10: a little picture/explanation could help.”
>
>     Thanks for this suggestion. We will add a figure and a short explanatory sentence to help the reader follow the definition.
>
> 3) “l225: and $p \in B_r^c$.”
>
>     Thanks for spotting this.
>
> 4) “l244: This is a bit fast here. I suppose the ReLU comes from the fact that the segments of the zonoid are based on 0.”
>
>     Yes, the rest follows from linearity. We will add a brief explanation to clarify this.
>
> 5) “l257: the [Carrière Bauer] paper mentioned in the ‘Contribution’ paragraph fits here?”
>
>     Yes, we will add the reference here.
>
> 6) “l273: this is a bit fast as well here. I assume that $Z_{\emptyset}=0$?”
>
>     Yes. We will make this explicit in Section 2.4.
>
> 7) “l477–483: some references should be added.”
>
>     Thanks for pointing this out. We will add references where appropriate and welcome further suggestions.
>
> 8) “l647: K.”
>
>     Thanks for noticing this.
>
> 9) “l777: By prop. 2, [...] to conclude with thm 1.”
>
>     Thanks for noticing this.
>
> 10) “l787: effective weighting.”
>
>     Thanks for noticing this.
>
> 11) “l793, 798: a factor 2 is missing.”
>
>     Thanks for noticing this.

---

> > ### Comment · Reviewer_4NTj · 2025-11-17
> >
> > Dear Authors,
> >
> > Thank you for your quick response. Using the same structure:
> >
> > Questions.
> > 1. "one can potentially have $M \to \infty$ without $d_1 \to 0$"
> > This is true indeed. But this bound is far from being sharp. For $R$-bounded diagrams:  $W_1 \lesssim \sqrt {R SW_1}$, c.f. [Theorem 2.1, arXiv.2510.16465]. So we have bi-continuity (and even Holder-continuity) on every compact, and in particular, for converging sequences of diagrams (as per your Def.6).
> > 2. From what I can read, [Bandiziol & De Marchi] only consider kernels. I agree this is reassuring, but I think it would still be interesting to compare with supervised learnable methods.
> > 3. oops, missed that indeed, Thanks.
> > 4. to 6. Ok.
> >
> > Point of consensus: Okay.
> >
> > I also agree with Reviewer YUj9 regarding the complexity experiemtns. The linear complexity is a **major** strength of this paper IMO, and should be empirically validated.

---

> > > ### Author Response · Authors · 2025-11-18
> > >
> > > We thank the reviewer for engaging in an interesting discussion.
> > > We provide here some further comments.
> > >
> > > 1) We agree with the reviewer that the result for the Sliced Wasserstein distance is very elegant and, in fact, we would like to prove an analogous result for PS in future work. Nevertheless, the convergence of a sequence of persistence diagrams does not imply that their supports are compact, or even bounded. For example, consider $D_n = ${$(x,y), (n, n + 1/n)$}$ \longrightarrow D = ${$ (x,y)$}, the diameter of the support of $D_n$ clearly diverges as $n \to \infty$.
> > >
> > > Similarly, the number of points in a convergent sequence of diagrams can be unbounded. For instance, $D_n = ${$(1, 1/n^2), \dots, (n, n + 1/n^2)$}$ \longrightarrow D = \emptyset$ in the 1-Wasserstein distance: the cost of matching all n points in $D_n$ to the diagonal is in fact 1/n.
> > >
> > > 2) We are currently preparing a runtime comparison of the computational complexity of obtaining the different topological summaries, along with additional simulations aimed at a sensitivity analysis of our weighting schemes. Regarding the inclusion of further benchmarks, for the time being we have decided to concentrate our efforts on adding a comparison with the “unsupervised” vectorization method suggested by Reviewer YUj9, namely Persistence Splines, for which we have ported the original Matlab code. While we certainly understand the desire to achieve the best possible performance in any data analysis task, we have already expressed our reservations about including a comparison with a “supervised” linearization method. Our main point is that these are tools designed for different purposes and, in our view, there is limited common ground for a meaningful comparison given the aim of the paper.
> > >
> > > We’ll explain ourselves.
> > >
> > > One would never resort to a "supervised" method when an "unsupervised" one is naturally the appropriate choice. E.g. in any small to moderate data setting, in situations where interpretability is important (note that all the parameters of the considered vectorization methods are highly interpretable w.r.t. the original diagrams), or in unsupervised scenarios.
> > >
> > > Conversely, if one is solely interested in optimizing predictive performance and has access to sufficiently large datasets, one will typically resort to "supervised" approaches, possibly overparameterizing the model and then relying on regularization and careful optimization procedures. In this perspective, if one were to introduce a new "supervised'' method, it would be reasonable to demand that it outperforms all available "unsupervised" alternatives (as in the PersLay paper); otherwise, its practical utility would be limited. This, however, is not the aim of our work.
> > >
> > > We note that also one of the latest comments by Reviewer YUj9 goes in this direction.
> > >
> > > To reinforce this point, observe that a "supervised" formulation of PS (for example, where the weight function is also learned, say via a spline decomposition, or by multiplication with a Gaussian mixture whose mixing weights, means, and variances are optimization parameters) would, given enough data, almost certainly achieve better predictive performance than the closed-formula version of PS that we introduce. Yet it is not clear what insight would be gained from documenting this. Similarly, taking another vectorization strategy, turning it into a "supervised" method by optimizing parameters that are otherwise fixed, and then comparing it to its original definition would yield very limited implications.
> > >
> > > In our view, it would be very interesting to consider all vectorization methods in their "supervised" incarnations, possibly including a representation learned in a purely supervised fashion (for example, via an autoencoder), and then compare their ease of optimization, predictive performance, and related aspects. Such a study would place the different approaches on a more conceptually homogeneous footing. But, again, this is outside of the purpose of our work.
> > >
> > > Finally, we emphasize that we have deliberately kept our downstream pipelines as simple and transparent as possible (vectorization followed by regression/SVM) precisely so that the comparison isolates and highlights the effect of the vectorization strategy itself, rather than conflating it with the choice of regression or classification model.

---

> ### Comment · Reviewer_4NTj · 2025-11-26
>
> Thank you for your detailed answer, and the updated version.
>
> 1. Ok, but in practice, practical datasets / filtrations are bounded with known bounds, and non-fractal (or "Morse-like"). So these bounds are usable in practice.
> 2. Ok.
>
> Regarding the comparison with other methods, I agree that it's not fair to ask for full performance / comparison with SOTA. But orders of magnitudes, with modern approaches, are interesting.
> Every supervised deep method to vectorize diagram is theoretically equivalent to the non-optimized versions, up to some parameter tuning, but not every method is easily learnt by a machine. Thus the remak about the SOTA comparison.
> Now, I don't understand the point/angle of the authors on the datasets:
>  - Interpretability. On small datasets, there are already plenty of unsupervised vectorization methods that “works” and, that I feel, are more interpretable. If the main interest is interpretability and not performance, why not choose a simple landscape (eventually tuned for the given task), or even some simple statistics on the diagrams?
>  - Overfitting. There exist datasets with fixed train and test splits such that overfitted methods perform bad (by design) on the test set. Why not compare your method on such datasets?
>
> Once again, I see the value in this contribution. This approach seems mathematically sound, with good performance (both statistically and computationally); but i feel that the comparison in terms of datasets and methods is a bit limited.
> Furthermore, this construction is at the expense of a vectorization that is a bit more involved than the Landscapes / Images, which doesn't help interpretability, so this has to be justified.
>
> Regarding the updated computational timings.
> I think Figure 7.f. should be a loglog scatterplot to clearly see the linear scaling. The timings look reassuring, though.

---

> ### Author Response · Authors · 2025-11-26
>
> Thanks again to Reviewer 4NTj for the discussion.
>
> 1) We agree that the bound is still useful and, in fact, we plan to investigate whether a similar result can be obtained for PSs.
>
> 2) Let us try to be as concise as possible and restate our position.
>
> A) On interpretability.
>
> We think we used the word "interpretability" with slight different meanings: interpretability of the representation and interpretability of the modelling choices.
>
> On the one hand, we believe that genuine interpretability of the representation is only possible at the level of PDs; once we switch to a continuous representation, a substantial amount of information is inevitably lost (at least visually). PLs, being a hybrid continuous–discrete representation, preserve more structure than fully continuous embeddings, but even in our experiments we had single PLs with over 1,500 functions. Focusing on the first landscapes can provide some limited insight, but truncating such a series has a very strong impact on the analysis, and interpreting only a handful of functions leads to rather questionable conclusions.
>
> On the other hand, when we fix, for example, the parameter K for PSs or the weighting function used for PIs, we are making an explicitly interpretable modelling choice: we decide that certain, well-identified parts of the diagram should be treated as less relevant (i.e., as noise). Moreover, the meaning of the parameters used to build PIs is clear: we are placing Gaussians at persistence pairs. If instead one optimizes over these parameters, one is effectively displacing the Gaussians in a way that is optimal with respect to the labels, but what kind of modelling choice does that correspond to at the level of the diagrams?
>
> That said, if one is training a deep neural network with many layers and parameters, it is indeed reasonable to also optimize the parameters of the vectorization for predictive performance.
>
> B) On “plenty of unsupervised vectorization methods.”
>
> We believe that having a variety of tools to tackle a problem is desirable, especially when the proposed additions are genuinely different and not strictly worse than what already exists. Our results clearly indicate that this is the case for PSs. The summaries we consider also differ significantly in the metric structures they induce on the space of PDs: PLs are not linear operators on the space of PDs (seen as measures), while PSs and PIs are. This can be beneficial or detrimental depending on the data. PSs have very high fidelity with respect to the 1-Wasserstein metric, which is why they tend to perform well in unsupervised settings (our clustering case study is based on 200 independent datasets) and on a fixed compact support, although working on the sphere makes them somewhat harder to handle. PIs, by contrast, are trickier to tune and we found them to perform slightly worse, on average, than the alternatives. PSpl are clearly very bad in preserving the metric structure of PDs, but they are well suited for supervised tasks.
>
> We also think that introducing tools for which theoretical guarantees were previously unavailable for similar functional representations is, in itself, a positive contribution. For example, motivated by our results on PSs, we are currently working on a bi-Lipschitz-type bound for PDs on a fixed compact support, leveraging the literature on Gaussian mixtures and using PDs as mixing measures.
>
> Finally, we do feel that the statement “the comparison is a bit limited” is somewhat harsh: in terms of methods that are direct competitors of PSs, we have tested a substantial number of them. Nonetheless, we understand that we are not in a position to insist too strongly, and we will make an effort to include a comparison with PersLay.
>
> (We will fix Figure 7.f)

---

### Official Review · Reviewer_D5QX · 2025-11-01

**Soundness:** 2
**Presentation:** 1
**Contribution:** 2
**Rating:** 0
**Confidence:** 3

**Summary:**

This paper presents persistence spheres, a new way to represent persistence diagrams as functions. The method provides a bi-continuous mapping that is Lipschitz continuous with respect to the 1-Wasserstein distance and has a continuous inverse on its image. These properties ensure both stability, meaning similar diagrams produce similar functions, and geometric fidelity, meaning that similar functions reflect similar diagrams. Together, they offer an optimal balance between robustness and faithfulness to the original geometry of persistence diagrams.

**Strengths:**

The experiments demonstrate that persistence spheres perform consistently well across a wide range of data types.

**Weaknesses:**

The paper as a whole reads more like a sequence of definitions and technical explanations than a cohesive narrative, which makes it difficult for readers to grasp the overarching motivation and significance of the work. While the theoretical foundations are clearly laid out, the presentation lacks a guiding storyline that connects these formal definitions to the broader research goals and practical implications. Nearly two pages are devoted solely to dataset descriptions, and almost another full page focuses on hyperparameter settings and implementation details. While such information supports reproducibility, it disrupts the flow of the paper and could be moved to an appendix or supplementary material. This space could instead be used to strengthen the narrative by clarifying research goals, providing interpretive discussion, and highlighting the significance of the findings. Moreover, several analytical gaps limit the depth of the study. There are no direct runtime benchmarks, so although computational efficiency is discussed qualitatively, explicit timing comparisons with existing topological representations such as persistence images, persistence landscapes, or the sliced Wasserstein kernel are missing. Hyperparameter sensitivity is not analyzed, leaving unclear how robust the method is to parameter changes, and no ablation studies are provided to isolate the effect of specific design choices such as the weighting function or spline grid size. Scalability testing is also limited to moderate-sized datasets, offering no insight into how the approach performs on large or high-dimensional data. Finally, the reported results rely on averages over runs but include no statistical significance tests or confidence intervals, making it difficult to assess whether performance differences are meaningful.

**Questions:**

How do persistence spheres advance the broader understanding of how topological information can be integrated into machine learning, beyond simply providing another competitive embedding for persistence diagrams?

---

> ### Author Response · Authors · 2025-11-12
>
> We thank the reviewer for the detailed feedback. Although, as reflected in the scores, our work did not receive much appreciation, we nonetheless wish to address several points and request clarifications that could help us improve the paper and better understand the motivation for the score received.
>
> We organize our response into two sections: QUESTIONS/COMMENTS (points for which we would appreciate clarification) and POINTS of CONSENSUS (points where we are aligned with the reviewer’s remarks).
>
>
> POINTS of CONSENSUS
>
>
> We thank the reviewer for pointing out that the choice of the parameter in our vectorization was insufficiently discussed. We fully agree. Since this parameter admits a clear interpretation, we will add a dedicated paragraph elaborating on its role.
>
> We anticipate that varying this parameter essentially corresponds to selecting a band along the diagonal that identifies points likely to represent noise, which are then down-weighted smoothly (via a Lipschitz step function). Hence, it could be chosen a priori when prior knowledge about the data is available.

---

> > ### Author Response · Authors · 2025-11-12
> >
> > QUESTIONS / COMMENTS
> >
> >
> > 1) On the length of the experimental section and the lack of statistical significance tests or confidence intervals.
> >
> >
> >     The reviewer notes that “nearly two pages are devoted solely…” and that “no statistical significance is reported.”
> >     We modeled the structure of our experimental section after standard practice in the field, as seen in conference papers introducing   the Persistence Scale Space kernel, the Persistence Weighted Gaussian kernel, the SWK, and the PerLay, among others. These works typically devote three pages to dataset descriptions and hyperparameter details, and report results in terms of average accuracy and standard deviation across independent runs.
> >
> >     Even in related journal papers (e.g., PI, PL, and Persistence Splines, a reference not mentioned in the paper but suggested by another reviewer), results are consistently summarized by means and standard deviations, without formal hypothesis testing or explicitly mentioning CIs, whose information is, however, basically contained in the mean-standard deviation metric.
> >
> >     We could include independent t-tests on our 10 repetitions; however, their interpretability is limited given the small sample size. As already mentioned, confidence intervals derived from the same statistics would provide nearly equivalent information to means and standard deviations. For these reasons, most prior works  follow the mean/standard deviation convention.
> >
> >     Hence, we would like to stress that the structural critiques raised here would apply equally to the cited literature, including journal publications. We therefore wonder whether it would be appropriate, or even desirable, for us to deviate from this well-established format.
> >
> >    2) On the lack of a “guiding storyline connecting formal definitions to broader research goals.” and "How do persistence spheres advance the broader understanding of how topological information can be integrated into machine learning, beyond simply providing another competitive embedding for persistence diagrams?"
> >
> >
> >     We believe the main goal of the paper is clearly stated and motivated in the introduction, which establishes a coherent narrative: we aim to introduce a new representation of PDs that provides stronger theoretical guarantees than existing alternatives, supported by illustrative applications and case studies. A broader discussion of why embedding topological information into machine learning models is useful lies beyond our scope, as this has been thoroughly addressed in prior literature.
> >
> >     Maybe, the exposition of our theoretical properties could be reinforced to make their relevance more explicit. For instance, consider a scenario where one aims to define a loss function based on a PD vectorization because computing Wasserstein distances is computationally expensive. In such a case, up to additional assumptions, persistence spheres would be the only representation guaranteeing that, if the loss converges to zero, the corresponding diagrams also converge. This is a nontrivial and practically meaningful property.
> >
> >     The current narrative was designed to mathematically justify the proposed approach. We could have omitted the lift zonoid construction and presented only the final explicit summary form, but this would have reduced mathematical clarity and and undermine the understanding (and recognition) of the theoretical foundations on which our method builds.
> >
> >     We would therefore appreciate more specific guidance from the reviewer on how the storyline could be improved or clarified, especially given the score we were assigned.

---

> > > ### Author Response · Authors · 2025-11-12
> > > **(continues the previous comment)**
> > >
> > > 3) On runtime comparisons.
> > >
> > >
> > >     We understand the origin of this critique, but at a closer look we believe that reporting overall cross-validation runtimes is of limited relevance in our setting, for a combination of several reasons:
> > >     (a) The methods compared are qualitatively different (kernel-based vs. explicit representations), making computational cost highly data set-dependent. For instance, it is trivial to construct datasets where kernel methods are disproportionately slower.
> > >     (b) Runtimes depend heavily also on the parameter grids explored during cross-validation, which can vary arbitrarily. For instance, for PLs we did not cross-validate the discretization grid or the number of landscapes, relying instead on reasonable prior choices, making it the fastest method. Conversely, PIs were much slower, due to a larger parameter set, despite other cross-validation simplifications.
> > >     (c) As noted in the paper, our current implementation is not optimized (e.g. we have an additional functional PCA step, fPCA, using spherical splines). As we write in the paper, the computational cost of our representation is comparable to evaluating a small neural network with a ReLU layer (of size (3,)) per point in the diagram, but, at the present stage, we then resort to fPCA to use common statistical learning packages. Optimized implementations, especially using sphere-specific statistical tools, will be developed as part of future work.
> > >
> > >     If desired, we could add a runtime comparison limited to evaluating explicit summaries (PS, PL, PI) on diagram populations of varying sizes and cardinalities. However, we expect these results to be of limited value, as all such representations are computationally efficient.

---

> > ### Comment · Reviewer_D5QX · 2025-11-26
> >
> > Thank you to the authors for the rebuttal. It addresses some of my earlier concerns and provides helpful clarification, so I am making a slight upward adjustment to my score.

---

### Official Review · Reviewer_FZZ6 · 2025-11-12

**Soundness:** 3
**Presentation:** 3
**Contribution:** 3
**Rating:** 6
**Confidence:** 2

**Summary:**

This paper proposes persistence spheres (PS), a new functional representation of persistence diagrams (PDs) for topological data analysis. Unlike existing embeddings, persistence spheres yield a bi-continuous embedding, which guarantees both stability and geometric fidelity. The paper defines persistence spheres via the support functions of lift zonoids of re-weighted persistence diagrams, proves continuity theorems, and provides explicit, efficiently computable formulas. Experiments across several benchmark datasets and show that persistence spheres are comeptitive or superior to established methods in regression and classification.

**Strengths:**

The paper demonstrates originality by introducing a mathematically rigorous yet computable embedding that enhances geometric fidelity compared with existing topological data analysis (TDA) vectorizations. Its quality is supported by solid theoretical foundations, complete proofs, and carefully reproducible experiments. The empirical validation spans diverse application domains, including functional, graph, mesh, and point-cloud data, underscoring the versatility of the proposed method. In terms of scalability, the approach achieves linear complexity with respect to the number of persistence diagram points and allows straightforward parallelization.

**Weaknesses:**

The paper lacks an ablation or sensitivity analysis on key hyperparameters such as the weighting function $\omega$ or the number of basis functions. These parameters likely influence both the expressiveness and stability of the representation, yet their impact is neither theoretically discussed nor empirically evaluated.
Grammar errors:
Line 465: “due the high variability” → should be “due to the high variability.”

**Questions:**

1. How sensitive are the results to the specific hyperparameter choices—particularly the weighting function $\omega$, the number of basis functions — and could the authors provide an ablation or sensitivity analysis to clarify their impact on performance and stability?
2. Runtime comparisons: the paper claims linear complexity, but actual wall-clock runtimes versus PIs/PLs are not reported—can authors provide them?

---

> ### Author Response · Authors · 2025-11-14
>
> We thank the reviewer for the feedback and agree on all the points raised, as already discussed in our replies to other reviewers.
>
> We will revise our paper accordingly.

---

### Author Response · Authors · 2025-11-26
**Revised Manuscript and Code**

We have submitted what we believe to be a significantly improved version of the paper.

We thank all reviewers for their valuable feedback, and genuinely appreciate those who engaged in the discussion phase.

Even Reviewer D5QX, for whom the content of the paper was not even enough to move the needle from the 0 score (and has refrained from discussing or explaining the inconsistencies in the provided review), has nevertheless (albeit marginally) contributed to improving the manuscript.

As requested by the reviewers, the revision primarily focuses on adding new benchmarks, case studies, and simulations, and on providing a detailed analytical and empirical study of the persistence spheres weighting functions and parameters.

Below we summarize the main changes, roughly in the order in which they appear in the manuscript:

1) Main contributions.
We slightly expanded the “Main Contributions” paragraph to highlight some implications of bi-continuity.

2) Lift zonoid illustration.
We added a figure to visually accompany and clarify the definition of the lift zonoid.

3) References after Corollary 1.
We now explicitly reference the works of Carrière and Bauer immediately after Corollary 1.

4) Persistence splines.
Following Reviewer YUj9’s suggestion, we included persistence splines in our comparison. To do so, we ported the original matlab code to python (providing a notebook with basic examples). This turned out to be an interesting and competitive addition to our benchmarks, particularly well suited to supervised tasks.

5) Clustering case study.
We added an unsupervised clustering case study to broaden the empirical evaluation.

6) Spherical harmonics vectorization.
We implemented the spherical harmonics–based vectorization of persistence spheres, using the pyshtools python package.

7) Random forest models.
We re-ran all pipelines (except SWK) using random forest regressors and classifiers, which proved much easier to tune. This also allowed us to successfully run analyses that previously failed to converge (e.g., persistence images on the ENZYMES JACC dataset).

8) Results table and confidence intervals.
We rewrote the Results section and the main results table to reflect the new benchmarks. In response to Reviewer D5QX’s concerns about confidence intervals (even though the necessary information was already encoded by mean, standard deviation, and number of independent runs), we now explicitly indicate which methods have confidence intervals overlapping with the top-performing method in each row.

9) Discussion.
We enriched the Discussion section with suitable references.

10) Introductory material on PDs (Supplementary).
We added to the Supplementary an introductory section on persistence diagrams, which is referenced in the main text.

11) Weight function analysis (Supplementary).
We included a detailed analysis of the weighting function and its parameters, combining analytic results on the geometry of the level sets with two simulation studies that illustrate their practical behaviour. We also discuss guidelines for exploring these parameters in cross-validation.

12) Computational aspects and runtimes (Supplementary).
All computational details have been moved to the Supplementary, where we additionally report a simulation study on runtimes.

13) Ablation studies (Supplementary).
We added ablation studies for all parameters used in persistence spheres, again collected in the Supplementary and referenced from the main text.

14) Typos and minor errors.
We carefully corrected all typos and minor issues pointed out by Reviewer 9Ldx.

---

> ### Comment · Reviewer_YUj9 · 2025-11-26
>
> The revision addresses most of my concerns. I have updated my rating.

---

> ### Comment · Reviewer_9Ldx · 2025-11-27
>
> I thank the authors for the updates. With new results for the unsupervised case the paper looks stronger. I also did not fully appreciate the computational advantage (complexity) of the method until reading the discussion. I will update my score.
> Another small point I wanted to make: if I understand correctly, the persistence spheres provide a _differentiable_ vectorization, not only in theory, but also in practice --- the formulas to go from PD to PS are very naturally expressed in PyTorch terms. So, if I want to define some loss in terms of the vectorization, torch will be able to back-propagate the gradient to the input. Differentiability of persistence diagrams has already been used in a number of papers, so this property may be worth mentioning in the final version. I am not saying that other descriptors do not have this property, of course.

---

### Author Response · Authors · 2025-11-27
**Revised Manuscript and Code 2.0**

In addition to the changes described in our previous comment, we have now included PersLay in the benchmark comparisons. We have also corrected the figure flagged by Reviewer D5QX. At this stage, we believe we have addressed all points raised by the reviewers, irrespective of whether we were fully convinced of their added value for our work.

---

### Meta-Review · Area_Chair_Rfcg · 2026-01-07

**Summary:**

Reviewers largely agree that this paper introduces a novel and theoretically well-grounded representation of persistence diagrams, distinguished by its bi-continuity with respect to the 1-Wasserstein distance and the existence of a continuous inverse (on its image)—a property that is valuable in topological machine learning. Theoretical contributions are viewed as sound and well-motivated, with clear advantages in stability, geometric fidelity, and parallelization. The main points of contention concern presentation and empirical scope: multiple reviewers find the exposition mathematically dense with insufficient intuition and visual explanations, while others request stronger empirical support through scale-up/runtime tests, hyperparameter ablations, and comparisons to more recent or supervised baselines. The authors’ revisions substantially strengthen the experimental section, leading several reviewers to raise their scores, though a minority remain unconvinced.

In my assessment, the strong reject (score 0; by one reviewer) is also not warranted, as the argumentation is unclear and imprecise (and I mostly agree with the authors remark(s) to the AC/SAC/Program chairs about the missing substance of that review which would substantiate the score) - I am therefore excluding this specific review from my assessment. Regarding the remaining "Reject" score, my assessment is that the authors provided a strong rebuttal which clarified most points and also that this revision would have raised the score of YUj9.

Overall, the paper is viewed as technically solid and original. I am therefore recommending Acceptance at this point, after carefully reading the original reviews, rebuttal and changes to the manuscript.

**Reviewer Concerns:**

Excluding the 0 score by one reviewer, I think the author clarified most concerns in their rebuttal. One might argue that the paper is technically too dense in parts and requires specific (TDA) prior-knowledge to fully grasp the contribution, however, this is true for most more theoretically oriented TDA+ML papers, or any other more theory-focussed paper for that matter.

**Reviewer Scores:**

In particular, I am convinced that YUj9 would have raised the score (at least to a "Marginally below ..." score) given the possibility to fully participate in the discussion with the authors (and given the rebuttal). Furthermore, I think some of the 6 scores would have gone up (to some extent), warranting acceptance in my point of view.

---

### Decision · Program_Chairs · 2026-01-26

Accept (Poster)